# Feasibility of integrating survivors of stroke into cardiac rehabilitation: A mixed methods pilot study

**Elizabeth W. Regan**[1]*, **Reed Handlery**[1¤], **Jill C. Stewart**[1☯], **Joseph L. Pearson**[2☯], **Sara Wilcox**[1☯], **Stacy Fritz**[1]

1 Department of Exercise Science, University of South Carolina, Columbia, South Carolina, United States of America, 2 Department of Health Promotion, Education and Behavior, University of South Carolina, Columbia, South Carolina, United States of America

☯ These authors contributed equally to this work.
¤ Current address: School of Physical Therapy, Arkansas Colleges of Health Education, Fort Smith, Arkansas, United States of America
* eregan@email.sc.edu

**Data Availability Statement:** All files are available from the Open Science Framework database (osf.io/cqm5p/).

## Abstract

### Background

Survivors of stroke are often deconditioned and have limited opportunities for exercise post-rehabilitation. Cardiac Rehabilitation (CR), a structured exercise program offered post-cardiac event in the United States (U.S.), may provide an opportunity for continued exercise.

The purpose of this study was to examine the feasibility of integrating survivors of stroke into an existing, hospital-based CR program through an assessment of (1) recruitment, uptake and retention, (2) adherence and fidelity, (3) acceptability and (4) safety.

### Methods

A mixed methods design combined a single group, pre-post design, pilot feasibility study with an imbedded qualitative inquiry. Survivors of stroke were recruited into a standard 12-week, 36 visit CR program.

### Results

Fifty-three survivors were referred, 29 started and 24 completed the program. Program uptake rate was 55% and completion rate was 83%. Eleven completers and one non-completer participated in the qualitative interviews. Program completers attended an average of 25.25 (SD 5.82) sessions with an average of 38.93 (SD 5.64) exercise minutes per session while reaching targeted rate of perceived exertion levels. Qualitative themes included perceived benefits of an individualized program in a group setting, positive interactions with qualified staff, opportunities for socialization, and regular monitoring and staff attentiveness promoting feelings of safety.

**Funding:** This work was supported by the University of South Carolina (USC) Behavioral-Biomedical Interface Program-NIGMS/NIH-T32 2T326M081740-11A1, bbip.sc.edu (ER); 2019 American Heart Association (AHA) Pre-Doctoral Fellowship, Heart.org (ER); 2018 American Physical Therapy Association Health Policy and Administration Research Grant aptahpa.org (ER); 2019 USC Support to Promote Advancement of Research and Creativity Grant, sc.edu (ER); AHA Grant 15SDG24970011, Heart.org (JS); Promotion of Doctoral Studies (PODS)–Level I Scholarship from the Foundation for Physical Therapy Research, foundation4pt.org (RH); the Arnold Fellowship from the Arnold School of Public Health, USC (RH). The funders had no role in study design, data collection and analysis, decision to publish, or preparation of the manuscript.

**Competing interests:** The authors have declared that no competing interests exist.

## Conclusions

Survivors of stroke were able to meet Medicare standard dosage (frequency and session duration) and rate of perceived intensity goals, and perceived the program as needed regardless of their mobility limitations or previous exercise experience. Primary challenges included managing referrals and uptake. Results support feasibility and benefit for survivors to integrate into U.S. CR programs.

## Introduction

More than 80 million people in the world are living after stroke with 13.7 million new cases each year [1]. Stroke is the leading cause of disability in the United States (U.S.) [2]. Impairments after stroke typically result in reduced physical activity which increases the risk for stroke recurrence and the development or worsening of comorbid health conditions [2]. Studies support the feasibility and safety of exercise training for survivors of stroke and suggest that engaging in exercise can improve cardiovascular risk factors and endurance while reducing disabilities [3–6]. Despite these known significant benefits, survivors of stroke face barriers to participating in regular physical activity (PA) due to limited self-efficacy, safety concerns, environmental restrictions, and lack of accessible community programs [7–9]. With a large number of survivors of stroke living with disability and at a higher risk for stroke reoccurrence and other diseases, there is an urgent need to reduce disability and modify cardiovascular risk factors [2, 10].

While some traditional rehabilitation activities can induce cardiovascular training effects, research has shown that during these programs, patients spend little time at the intensity levels required for endurance changes: only 24% of time at > 40% maximum heart rate (MHR) in one study [11], and 4.8% of time at > 60% MHR in another [11, 12]. Rehabilitation stays in the U.S. are declining in length due to reduced insurance reimbursement [13], potentially compounding the deconditioning remaining when rehabilitation is complete [14, 15], The lack of appropriate exercise programs available for survivors impedes continuation of supervised activity after rehabilitation [16]. Most survivors do not continue exercise or participate in PA after formal rehabilitation. Daily step counts for community-dwelling survivors are commonly less than 3000, well below a 6025 step cutoff for predicting new vascular events [17–19].

Cardiac rehabilitation (CR) programs are an essential component of recovery after cardiac events [20]. CR programs improve participants' health through cardiovascular endurance activities, resistance and stretching exercises, educational programs, and stress reduction efforts [21]. These programs are widely available, staffed by experienced health care professionals, and are well-established in the medical infrastructure [22, 23].

Research studies, primarily outside of the U.S., have implemented CR programs exclusively for survivors of stroke and included survivors of stroke within cardiac-diagnosis specific programs [24–29]. One example is a program in Canada that provided a stroke-specific program for survivors of stroke with remaining mobility deficits post-rehabilitation and integrated those without mobility deficits into a traditional CR program [29]. The stroke-specific program was a once weekly 90 minute class including aerobic exercise, resistance training and health education [29]. Another program in the United Kingdom was a six-week program including 60-minutes of exercise, 60-minutes of education and 10-minutes of relaxation for participants after transient ischemic attack (TIA) [26]. While these studies support feasibility and benefit for survivors of stroke, several limitations exist. Limitations in existing studies

include variation in dosage, limited evaluation of participant perception and limited external validity [3–5]. Additionally, it is difficult to apply these findings to U.S. settings due to differences in dosage, costs, and insurance coverage. As a result, a knowledge gap exists for the feasibility of CR for survivors of stroke within U.S. programs that follow Medicare guidelines [30]. U.S. Medicare guidelines require CR programs to include exercise, risk modification intervention, psychosocial evaluation, and outcomes assessment within an individualized patient plan [30]. A maximum of 36 sessions within 36 weeks is the basic allowance, with at least 31 minutes of active exercise required each session [30].

The purpose of this pilot intervention study was to examine the feasibility of integrating survivors of stroke into an existing, outpatient CR program in the southeastern U.S. through an assessment of (1) recruitment, uptake and retention, (2) adherence and fidelity, (3) acceptability, and (4) safety [31–33].

## Materials and methods

The study was conducted at an outpatient CR site affiliated with a U.S. multi-hospital health system in the state of North Carolina from August of 2018 –November 2019. A mixed methods design combined a single group, pre-post design, pilot feasibility study with a pragmatic, qualitative inquiry of participant perception [32, 34]. The study was a registered clinical trial through the United States National Library of Medicine (ClinicalTrials.gov ID: NCT03706105). The health system Institutional Review Board (IRB) approved this study (18–1001), and the University of South Carolina IRB (Pro00079131) acknowledged it. All data and materials have been made publicly available at Open Science Framework [35]. The health system CR program had an existing protocol for non-Medicare covered diagnoses, primarily people with heart failure and cancer, to participate in the program modeled using a 12 -week structured and medically monitored outpatient CR protocol [30].

### Participant recruitment and eligibility

Hospital system providers (medical and rehabilitative), community support groups, and word of mouth recruited survivors of stroke for this 12-week, 36 session CR program.

The following inclusion criteria determined eligibility for participation in the study:

1. a diagnosis of stroke at least 3 months prior to reduce the likelihood of outcomes being confounded by spontaneous recovery;

2. completion of physical and occupational therapy rehabilitation, if applicable to avoid confounding the results;

3. clearance by treating medical provider (physician or nurse practitioner) to participate as a standard CR program requirement;

4. ability to walk at least 40 meters with or without an assistive device as a standard CR program requirement;

5. ability to transfer from sit to stand without external assistance as a standard CR program requirement; and

6. ability to follow instructions and to communicate exertion, pain and distress as a safety measure.

Potential participants were excluded from the study for any of the following:

1. acute medical problem rendering exercise unsafe;

2. significant pain that prevented standing or interferes with movement; or

3. history of additional, non-stroke, neurologic condition.

## Study procedures

Consecutive sampling from referral sources and community interest identified potential participants. Referrals from health system sources used standard referral procedures through electronic medical records. Outside the health system referrals were accepted from physicians. Once initial eligibility and interest were determined, participants completed an evaluation at the CR site. The evaluation included informed consent, a physical therapy screen, basic demographic data, and a battery of outcome measures. The primary investigator (PI), a licensed physical therapist, performed the screen to verify eligibility and determine any modifications (i.e., equipment use, movement limitations, etc.) required for CR equipment or activities. Efficacy of the program was assessed and presented separately [36].

The physical therapy screen and outcomes informed equipment or activity modifications or cautions within the standard CR program which were documented in a plan of care shared with CR staff. The only differences between the standard CR program and the study program were (1) Individual equipment or activity modification recommendations, and (2) the availability of the PI for consultation (in person and by phone) to address any mobility or safety issues arising.

## Standard program description

The standard exercise program began with an analysis by the Exercise Physiologist (EP) to determine baseline levels of exercise intensity in metabolic equivalents (METs) based on participant's 6MWT results [37, 38]. Target heart rate (HR) was estimated from resting HR and HR at the completion of the 6MWT at a level between 60–85% of heart rate reserve. Borg rating of perceived exertion (RPE) goals were set from 11–14 (somewhat hard to hard) on the 6 (no exertion) to 20 (maximum exertion) scale [38, 39]. Standard program sessions were scheduled three times a week for 12 weeks with a target of 31–50 minutes of cardiovascular endurance activity each session. All standard program participants chose their days and times based on their schedule and program availability (Monday, Wednesday, Thursday 6am– 6pm, Friday 6am-12noon). Training sessions were individualized for all standard program participants. While components varied by session and individual, the standard format was 8 to 10 minutes of warm up, 10 to 40 minutes of cardiovascular endurance activities (treadmill, recumbent step machine, recumbent bike, over ground walking), and a 5 to 10 minute cool down followed by optional activities. HR and RPE were recorded for each endurance activity during the session. Optional activities within the standard program included strengthening, stretching, and/or relaxation, depending on the individual needs and goals. Progression in the standard program was determined through participant response. If RPE was consistently rated $\leq 11$, METs level effort was increased to reach an RPE of 14. Regular standard monitoring was completed both pre- and post-session for blood pressure, HR, and as needed, blood sugar and heart rhythms. Discontinuation of a standard session occurred if blood pressure exceeded 170/100 or by clinical expertise of the EP or staff nurses.

Psychosocial and nutritional consultation were available to all standard participants and were offered to study participants with stroke as but were not required. The PI discussed these options with study participants at the initial evaluation, and study participants were instructed to discuss their interest with the primary EP. Interest was recorded on the plan of care.

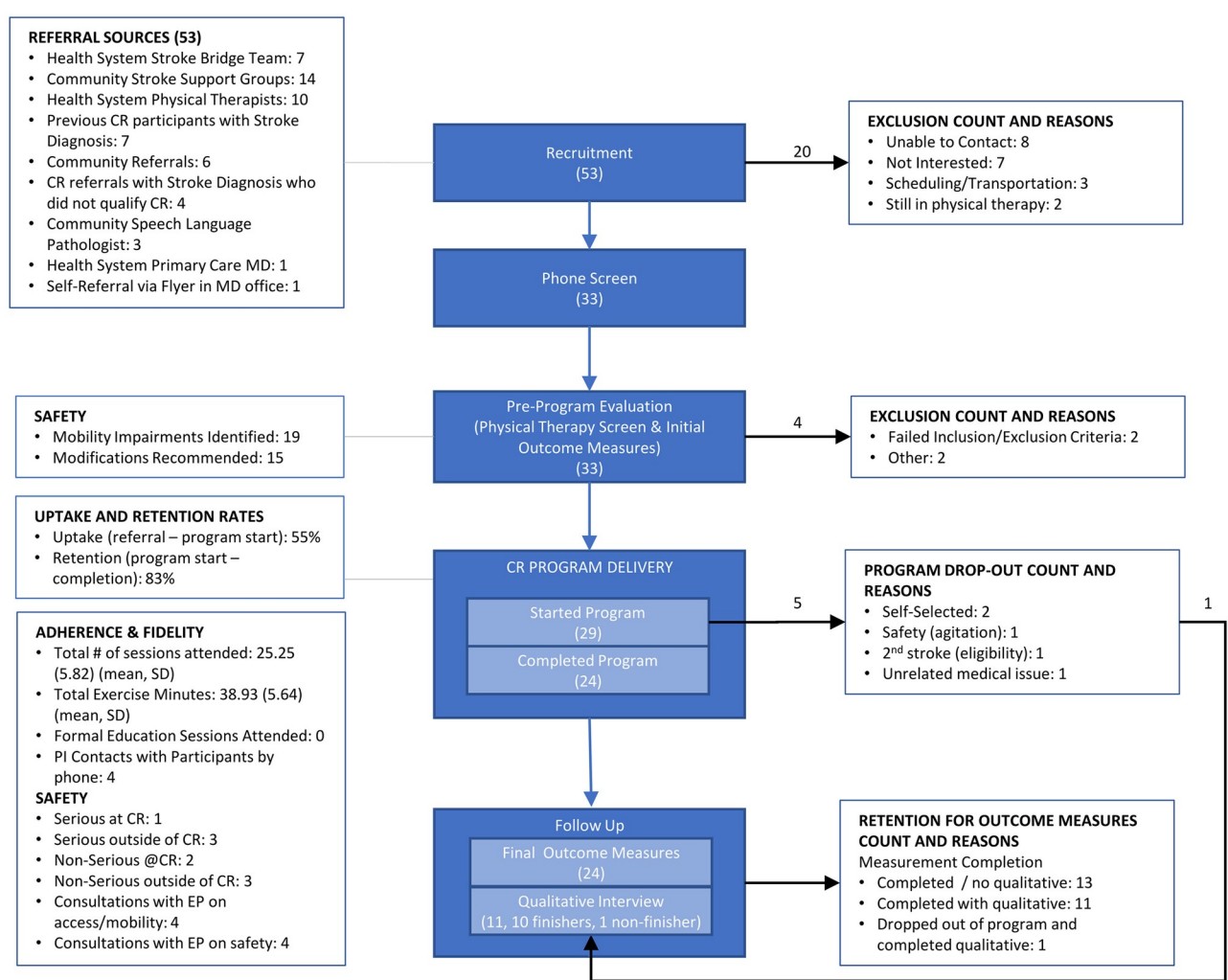

**Fig 1. Flow diagram of study with feasibility outcomes.** Abbreviation: CR, Cardiac Rehabilitation; MD, Medical Doctor; SD, Standard Deviation.

The CR program described was free for study participants; the per participant cost ($237) was covered by the study. At the end of the 12-week CR program, all participants who began the program were re-assessed using the study outcome measures.

Process variables and feasibility measures were recorded and analyzed throughout the study (Fig 1). The following categories were analyzed for feasibility, with details on which areas were covered by quantitative or qualitative measures in Table 1: (1) recruitment, uptake and retention, (2) adherence and fidelity, (3) acceptability, and (4) safety.

## Feasibility quantitative measures and analysis

Demographic data was collected on all participants starting the program using a standardized intake form including age, gender, race/ethnicity, date of last stroke, stroke type, work status and exercise frequency.

Details of the quantitative process variables are presented in Table 1 and include measures for recruitment, uptake and retention, adherence and fidelity and safety. Descriptive statistics for program intensity fidelity included means and standard deviations for minimum and maximum HR, minimum and maximum target HR, and percentage of time below, in, and above

**Table 1. Feasibility measures and qualitative interview topics.**

| Topic | Feasibility Measures | Qualitative Interview Topics |
|---|---|---|
| Recruitment, Uptake and Retention | • Number of referrals from each source<br>• Number phone screened<br>• Number evaluated<br>• Number eligible to participate, number eligible refusing participation,<br>• Descriptions of limitations<br>• Number completing program<br>• Number dropping out of program<br>• Number completing qualitative interviews<br>• Uptake rate (recruitment to program start)<br>• Program completion rate (start to completion) | • Recruitment (source, initial motivation for attending)<br>• Participation Barriers<br>• Participation Facilitators |
| Adherence and Fidelity | • Average number of sessions<br>• Total completing at least 18 of 36 visits [a]<br>• Average nutrition and exercise consultations<br>• Number attending weekly education sessions<br>• Number consulting psychologist<br>• Frequency of each exercise activity (% of sessions)<br>• Frequency of optional activity (% of sessions)<br>• Average session exercise minutes ($> 31$ minutes) [a]<br>• Average exercise minutes spent at target intensities<br>• Number of PI-participant consultations | • Participation Barriers<br>• Participation Facilitators |
| Acceptability | | • Capability, components, dosing<br>• Relationships (staff, other participants)<br>• Modification recommendations and preferences |
| Safety [b] | • Number and type of serious and non-serious events<br>• Mobility and safety consultations | • Factors that promoted safety<br>• Participant's perception of their safety |

[a] Minimum standard for Medicare guidelines of cardiac rehabilitation [30].

[b] Serious safety events were defined as any injury or medical issue requiring absence $>$ one week from the program

target heart rate ranges. Intensity fidelity measures also included calculations for median minimum and maximum RPE for each session, each participant and the entire sample.

## Qualitative procedures and analysis

A pragmatic, qualitative approach described participant perspectives [32, 34]. Table 1 presents key areas of evaluation based on a previous study evaluating CR for colorectal cancer [40] and contributing to participant perspectives on key pilot feasibility study characteristics including recruitment, uptake, retention, acceptability and safety [32] Participants' perceptions on program benefits are presented elsewhere [41]. Criterion sampling was utilized; participants who had previously participated in CR or had verbal communication limitations were excluded from the qualitative portion. All others who had started the program were invited to participate. Those participating completed informed consent and received a $20 gift card in appreciation of participation. Semi-structured interviews were conducted by the study PI after participants completed or left the program in person in a private room at the CR site. The study PI has been trained and mentored in qualitative methods as part of her doctoral training and completed previous published qualitative work. Interviews were recorded and transcribed verbatim. The number of participant interviews was determined by voluntary participation. In

addition to interviews, the PI completed monthly structured observations of participant activities, interactions with others, and barriers and facilitators including safety to provide supplementary data and to add to qualitative rigor [42].

Deductive and inductive thematic analysis were completed using NVivo software (version 12, QSR International) [43]. The PI completed the first round of open coding using in vivo style to stay close to participant phrasing; initial coding was completed for all transcripts and structured observations [43]. A codebook was created with broad categories based on inductive findings and feasibility topics (Table 1) for recruitment, barriers and facilitators, program delivery (safety, gym environment, interaction with staff, dosing, socialization, activity preferences, adherence, and recommendations for changes to program), and other. Two researchers (the PI and second author) iteratively completed deductive coding using the codebook in three rounds. Discussions between coders and with research mentor (last author) supported refinement and attempted to reduce bias. Data conflicting with primary themes were highlighted to present alternative viewpoints [43]. Results were reviewed, codes refined and finalized with a researchers and mentor.

## Results and discussion

A flow diagram presents a summary of study flow and feasibility findings (Fig 1). The study recruitment and participation period lasted for 15 months from August 2018 through November 2019. A total of 29 participants began the program, 24 completed the program (attended at least part of the program with final outcome measures available [44]). Demographic details are presented in Table 2.

Eleven completers and one non-completer participated in the qualitative interviews. Of the remaining 13 completers, 10 did not qualify for qualitative inclusion, one declined, one had unusable audio, and one left the country after program completion. The remaining non-completers either did not qualify or were unable to be reached.

Table 2.  Demographic details of cardiac rehabilitation program completers (n = 24) and non-completers (n = 5).

| Gender, % (number) | Age, mean (SD), range | Race / Ethnicity, % (number) | Time Since Stroke, mean (SD), range | Work Status, % (number) | Walk Aid, % (number) | Initial 6MWT Distance, mean (SD) | Initial Self-Selected Walking Speed, mean (SD) | Pre-Program Exercise Level, % (number)* |
|---|---|---|---|---|---|---|---|---|
| | | **Completers** | | | | | | |
| 79% (19) Male 21% (5) Female | 62.2 (12.4), 33–81 years | 71% (17) White 25% (6) African American 4% (1) Asian | 29.7 (29.9), 3–123 Months | 8% (2) Full Time 42% (10) Part Time 50% (12) Not Working | 75% (18) None 21% (5) Single Point Cane 4% (1) Hemi-Walker | 397.8 (119.2) Meters | 1.17 (0.21) m/s | 12.5% (3) None 12.5% (3) <1 x week 37.5% (9) 1–3 x week 37.5% (9) > 3 x week |
| | | **Non-Completers** | | | | | | |
| 60% (3) Male 40% (2) Female | 68.4 (15.0), 55–89 years | 60% (3) White 40% (2) African American | 37.0 (41.8), 13–109 Months | 20% (1) Part Time 80% (4) Not Working | 60% (3) None 20% (1) Single Point Cane 20% (1) Walker | 279.7 (147.1) Meters | 0.67 (0.28) m/s | 40% (2) None 0% (0) <1 x week 60% (3) 1–3 x week 0% (0) > 3 x week |

* Pre-Program exercise level was self-reported by participants. Abbreviation: SD, standard deviation; 6MWT, Six-Minute Walk Test; m/s, meters per second. Survivors of stroke with self-selected walking speed of greater than 0.8 m/s are considered unlimited community ambulators [45].

### Recruitment, program uptake, and retention

**Recruitment.** Over a 12-month recruiting period, 53 potential participants were referred. The largest number of referrals came from local stroke survivor support groups (n = 14). The PI provided education on the benefits of post-stroke exercise and the details of the program at support group meetings. Clinicians (rehabilitation providers, nurses and physicians), CR staff, and community referrals provided the remaining referrals (Fig 1).

Qualitative responses revealed participants found out about the study because of a local support group (n = 5), through a health system medical provider (n = 3) or through a community contact (n = 4). Participants were initially motivated to join the program because of desire for structured exercise, goals for health or symptom improvement, and altruism to support other stroke survivors and the researcher.

> Participant 3: "Well, I remember what you said in the presentation about, um, stroke survivors do not do enough aerobics. And um, I have never been enthralled by aerobics (laughing). And I thought this might just be the time for me to check it out."

> Participant 5: "Well, I was initially coming because I was trying to build up my stamina and everything because I had a long-term goal. The long-term goal is in September to go to [foreign country]."

**Uptake.** Program uptake rate (referral to start of program) was 55% and completion rate (start to finish of program) was 83%. Twenty referrals did not move forward to phone screen because of inability to contact, disinterest, conflicts or ineligibility (Fig 1). All of the 33 referrals who were phone screened advanced to initial participant evaluation. Four participants did not start the program: two failed the in-person screening (pain with movement, dependent with transfers) and two qualified but declined after the participant evaluation (scheduling conflicts, problem with physician referral).

Twenty-nine participants began the program. Most walked unaided (n = 21), with the remaining using a single point cane (n = 6) or a hemi-walker /walker (n = 2). Participants described remaining deficits related to their stroke as weakness (n = 13), walking (n = 10), balance (n = 9), coordination (n = 7), speech (n = 6), cognitive (n = 5), vision (n = 4) and memory (n = 4). Five participants reported single deficits: weakness, balance, speech, memory and sensation. The remaining 19 participants reported multiple deficits. Fifteen participants received recommendations for modifications to the CR program. Recommendations were primarily related to limitations or cautions on treadmill use or track-walking with an assistive device only. There were a few limitations of upper extremity overhead motion due to shoulder pain. Other recommendations were safety related, such as expressive speech limitations requiring visual cues for use of the RPE scale, and the presence of orthostatic hypotension requiring slow transitions.

Five participants did not complete the program either by choice, due to medical complications, or both (Fig 1). Two self-selected to discontinue the program; one participant who completed two sessions cited transportation and other medical issues while another participant who completed nine sessions cited increasing headaches, knee pain and medical uncertainty (Fig 1). Three other participants did not complete the program due to safety concerns related to cognitive issues, ineligibility after second stroke, and complications due to recurrent bronchitis; these participants completed one, 11 and 23 sessions, respectively (Fig 1).

**Retention.** Barriers to starting the program are listed above in the Uptake section. The remaining barriers and facilitators to continuing in the program and participating regularly were identified by the qualitative responses. Barriers included medical complications, competing time demands, financial concerns, transportation difficulties (including long distance to site), and disinterest in exercise. Facilitators to study participation included availability of social support, perceived benefits of exercise, intrinsic motivation and sense of commitment, and ease of transportation.

*Medical complications*: Medical complications impacted a few of the participants' attendance in the program (42%, n = 5). Perceived impairments impacted specific activities, minor illness or sleep disruption caused missed sessions, and for one participant, significant knee pain and headaches caused him to leave the program.

Participant 15: "Yes. I ah, ah. . .two or three times [missed sessions]. I had bouts of coughing at night and ah, um not being able to stop coughing ah resulting in not sleeping and ah, not going to work the next day or, and in a couple cases ah, um missed a couple days. Um, vomiting and ah, um just being tired probably."

Participant 25 (non-completer): "So it [headache and knee pain symptoms] was makin' my work out here more difficult. Even though I would puff through it, umm, it was still more overbearing for me than, I probably shouldn't have done it but."

*Competing time demands*: Participants (75%, n = 9) cited other life demands such as work, complications in home life, and travel for holidays and vacations as impacting session participation over the 12-week period.

Participant 4: "So my, not being able to here three times a week um, work came into play."

Participant 11: "I've had glitches where I like missed a day because of chaos in my personal life."

*Financial concerns*: While most participants did not mention financial concerns, the no-cost factor facilitated three (25%) enrolling in the study. One participant noted that alternatives such as personal training were too expensive. Additionally, two participants (17%) cited financial concerns as barriers to continuing to participate as self-pay clients at CR after the study ended.

*Transportation difficulties including long distance to site*: Two participants (17%) cited distance from their house as being difficult. Another had some limitations in driving due to vision loss and did not like to drive in the rain.

*Disinterest in exercise*: Participant comments revealed that for four participants (33%), gym machine exercise was not their preferred activity. One participant preferred riding her horse or dancing, which she perceived as more fun. Another had never exercised regularly in a gym and had to adjust at the beginning of the program. The other two simply did not like to exercise at all. All participants overcame barriers and believed exercise was important to their health.

Participant 16: "It's somewhat monotonous, and I don't like feeling fatigued and uncomfortable and tired."

Interviewer: "But you do it anyway?"

Participant 16: "I do it anyway, and I do feel better when it's all over."

*Availability of social support*: A majority of participants (75%) noted social support as a facilitator to their program participation. Whether that be from a spouse, family member, staff, or other CR participants, having someone to encourage them and notice their progress was a key facilitator.

Participant 11: "Yes, my husband is supportive. He wants me to continue doing it, he says that he's seen tremendous change and tremendous improvement. So, um you know, I maybe don't see it or feel it as much because I'm the one participating, but he, and he said that in terms of my stamina and overall like, you know."

Participant 19: "You know, if you get alone, and maybe somebody who's paying a little bit more attention to you, then you think they're paying attention to somebody else because they like to talk, or the subjects that you talk about are interesting to them and yourself. That motivates you to show up. "Oh, I'm gonna see [staff EP] today because we're talking [topics of mutual interest]. Well, so, you know, we have, we've always had interesting conversations. So that motivates you to come."

*Perceived benefits of exercise*: Seven participants (58%) acknowledged that exercise was important to their health. Six (50%) also either had previous positive experiences with exercise or noted program results contributed to their on-going participation.

Participant 5: "Because I know that exercise is very, very beneficial. I know that. I just have to be motivated to do it that's all (laughing)."

Participant 16: "Because I. . . I had a stroke and I don't want that to happen again. And everyone says exercise is good."

*Intrinsic motivation and sense of commitment*: Most participants (75%) noted either an ability to push themselves towards their goals and/or a sense of following through on obligations that helped facilitate their on-going participation. They felt accountable to themselves, to the staff at CR, and to the study PI.

Participant 5: "I um am kinda a little bit self-motivated, but then when you get here you get extra motivation too."

Participant 11: "Well my personality is such that if I commit to do something, I'm gonna do it. Even if it sucks, even if I hate it, even if I feel terrible, I'm gonna. I, I, my life is the suck it up principle. You suck it up and you [expletive] do it.

Participant 19: "So there was an accountability to myself, accountability to the trainers that are here, and accountability to [PI]."

*Ease of transportation or close distance to site*: Living or working close to the CR facility was a facilitator for four (33%) of the participants. Being close was convenient for participants, which made it easier to fit in sessions with their other obligations. Being nearby also allowed one participant to drive to the site even though she was uneasy driving, and another participant was able to walk to sessions, easing his burden of getting rides from friends and family.

Participant 3: "It was very easy for me to get here, you know its ten minutes away. And um once I found the place, I could get here very easily. Um, and so that was good. I've driven myself which is very odd for me. I drive very little and only in the daytime and you know only in the neighborhood."

### Adherence and fidelity

HR and RPE targets, HR averages and RPE medians are presented in Table 3 and Figs 2 and 3. While HR goals were not met, the calculations included warm-up and cooldown periods which were not intended to be in the target range. While medications may blunt HR response, RPE provides another measure of intensity, and RPE target goals were met. Twenty-one participants (87.5%) completed at least 18 sessions, with the remaining three completing 12, 17, and 17 sessions respectively. The participant who completed only 12 sessions cited demand for work as a barrier to more regular attendance. No study participants attended available and optional weekly education sessions or psychologist consultation. Education of participants was informal, depended on the staff member and the participant, and included nutrition and exercise consultation and advice. Staff noted on exercise logs discussions based on participant request for information or just general information sharing from supervising staff (EP, nurse, dietitian) during exercise sessions and recorded in notes on exercise logs. Detailed nutritional plans were sometimes provided. Both exercise and nutrition consultations often included an accountability component where participants were asked about their home food intake or exercise. Nutrition consults occurred on average of 2.04 (SD 1.52) times per participant (range 0–5 times), and exercise consultations occurred on average of 1.54 (SD 1.47) times per participant (range 0–4 times).

Walking on the indoor track was the most common session activity (91.42% of sessions) and often was the warmup and/or cool down activity. Other equipment included in sessions were recumbent stepper (67.49% of sessions), upper body ergometer (48.68%), aerodyne bike

Table 3. Target and actual aerobic exercise minutes, rating of perceived exertion and heart rate ranges.

| Measure | Target | Session Results, Mean (SD) or Median (IQR)[a] | Session Results, Range |
|---|---|---|---|
| Minutes of Aerobic exercise | > 31 minutes total: <br> 8–10 minutes warm up <br> 10–40 minutes moderate activity 5–10 minutes cooldown | 38.93 (5.64) minutes total | 29.25–51.41 minutes total |
| RPE Minimum | 9 (warm up and cooldown) to 11 (moderate activity goal) | 11 (0.625)[a] | 8–15 |
| RPE Maximum | 11 (warm up and cooldown) to 14 (moderate activity goal) | 13 (1.000)[a] | 11–16 |
| HR Session Minimum, BPM | 97.74 (12.16) | 90.57 (10.06) | 65.92–103.5 |
| HR Session Maximum, BPM | 115.14 (11.71) | 108.45 (12.02) | 75.42–129.19 |
| % of session time in or above target HR range | | 57.63 (27.36)[b] | 13.28–98.87 |
| % of session time below target HR range | | 38.91 (25.18) | |
| % of session time unrecorded HR range | | 3.46 (7.58) | |

Abbreviation: SD, standard deviation; IQR, interquartile range; RPE, rating of perceived exertion; HR, heart rate; BPM, beats per minute.

[a] Median (IQR)

[b] Total time includes warm up and cool down times which are intended to be lower than target ranges.

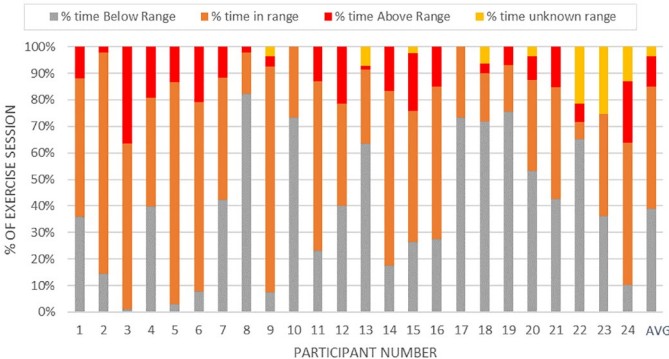

**Fig 2. Heart rate fidelity: Average of total session time spent below, at, or above target heart rate ranges for each participant.**

(41.09%), recumbent bike (39.77%), treadmill (37.79%), and elliptical (36.96%). Optional exercise components offered after regular exercise sessions included weekly guided relaxation, chair strengthening exercises and yoga. Relaxation was the only optional session recorded on exercise logs and averaged 2.71 (SD 2.42) times per participant or 10.73% of sessions.

Opinions on the types of exercise equipment they liked and did not like varied widely among participants. Six participants (50%) liked the recumbent or upright bike or the recumbent stepper. The most disliked pieces of equipment were the elliptical and the upper body ergometer/arm bike (50%). Participants highlighted their enjoyment of the variety of machines, being encouraged to try several different types and having some influence on what equipment to use regularly. Reasoning for preferences were due to enjoyment of the equipment, their perceptions of their capabilities on the equipment or personal goals to work a body area that was the equipment focus.

Participant 15: "[I liked] machines that ah involved my legs ah I. . .bicycle for all my life and uh so yeah, it was good to get back on something that uh either reclining bike or upright bikes.

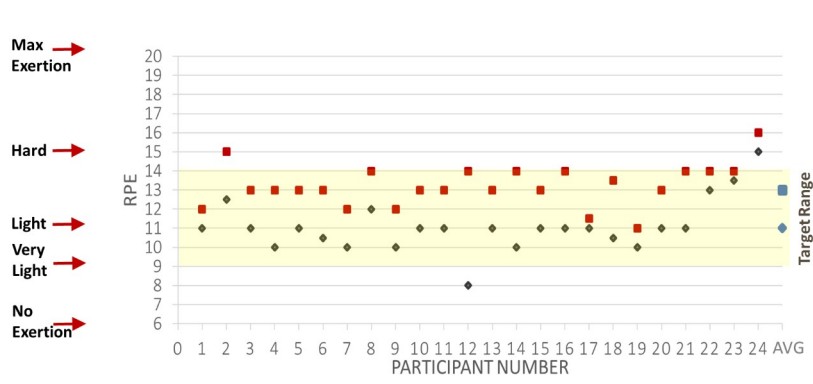

**Fig 3. Session minimum and maximum rating of perceived exertion medians.** Abbreviation: RPE, rating of perceived exertion. Shaded area indicates target RPE zone, warm up or cool down between 9–11 (very light to light) and moderate intensity activity between 11 (light) to 14 (somewhat hard).

Participant 11: "I mean there are a couple machines that I knew I did not like to use and that did not work for me. And they were nice about saying, 'okay well you don't have to'. They made me try everything, but there were ones that I just was not going to continue using."

Relaxation was mentioned by four (33%) of the participants as an enjoyable and positive aspect of the program.

Participant 3: "I really liked it. I attempted to do relaxation on my own but not been so successful. Um, so the group setting where everybody is quiet, and eyes are closed and there is this very gentle voice leading us through. Um, was very very good. And um, I felt like um you know, I didn't go to sleep but I felt sort of drowsy. And then when she brought us back into the present um, I just felt peaceful. It was good."

Non-safety related consultations by PI with participants came at the request of the primary EP. There were four participants who were absent more than two weeks and the EP requested PI contact by phone to determine concerns and encourage return. Of these four, one returned to the program and the other three left the program.

## Acceptability

Program acceptability was evaluated using qualitative responses. Resultant themes included the perceived benefits of an individualized program in a group environment, correct dosing with a desire for more scheduling options, positive interactions with staff who were qualified, and a supportive, energetic environment with opportunities for socialization and connection.

*Perceived benefits of an individualized program in a group setting*: Participants noted that although there was structure to the program, they were able to customize it to their abilities, interests and plan. Participants were individually monitored and encouraged to challenge themselves. Staff modified activities or provided support when necessary, such as using straps to support a weak limb on the recumbent stepper and assisting another participant who wanted to work on strengthening with the leg press machine.

Participant 4: I'm, I'm gonna come to say (pause) for the most part it was the right level. And that's another thing that your staff was doing, is, is no one was pushing anyone to do anything that they were not comfortable with. And uh, you also, I mean, I, I'd be asked what would you like to do next? Where would you like, you know, what exercise would you prefer? Where would you like to be? And so, so it's pretty much left to the individual and I shouldn't be speaking for everybody else. So for me, I did what I was comfortable with."

Participant 11: "Well, um I think it was really good, I really liked it. I liked that once I figured out what I was supposed to do, I could kind, it was kind of self-guided. You know, I was monitored but I could kind of control what I was doing. And liked that I wasn't um, I could kind of zone out and do my own thing."

*Acceptable frequency with a desire for more exercise day options*: All participants thought that three times weekly was an appropriate frequency. All except one participant thought the 12-week duration was also appropriate; one wished it was longer because of the benefits she was seeing. Three participants (25%) noted that they would prefer Tuesday or Friday afternoon options to get in three sessions a week.

Participant 5: "Three times a week is good. Um, the other thing I would do is every other day, Monday, Wednesday, Friday."

Participant 25 (non-completer): "Three, three times for an hour is like, like, no big deal."

Participant 16: "It was nice in that I could see a finish line. You know I was going three times a week, working out hard but I knew there was an end point and I'm going to continue to workout, I'm going to go to the Y, maybe just call it a milestone rather than an endpoint."

*Positive Interactions with Staff who were qualified*: Participants reported regular, repeated, appropriate interaction with staff throughout their sessions.

Participant 3: "I think the individual attention here is as good as the one on one stuff at the hospital [rehabilitation]. Because whoever was assigned to me would get me started with, set the machine and time me and they would almost to a person would come back at exactly the right time and ask how it went. And then do all the, you know, how hard was it? I really felt cared for."

Participant 15: "Ah, [nurse] was ah attentive. Ah, that's a. . .they all were if they took up the slack if uh, uh, uh [Primary EP] wasn't available. [Nurse 2] was very helpful and uh, um, uh and remembering what ah was my particular uh weakness and so on."

Participants described the staff as encouraging, caring, and enthusiastic. There was a team approach to supporting the participants. Study participants regularly mentioned the primary EP had a fun and energetic personality but also was skilled and attentive to their efforts and exercise responses.

Participant 19: "If there is 10 trainers here at one time, everybody helps everybody. So it isn't, you know, "Just wait for [primary EP], I can't help you." The next person would recognize that, okay, this guy's done or are you okay? Constantly being checked on by all of the team, and if I needed help, I wasn't afraid to ask then, you know, somebody else other than [primary EP], because at other times [primary EP] was in, into, involved with, you know, helping somebody else. So, very team oriented in that sense.

Participant 3: "I thought [head exercise physiologist] was particularly skilled in reading me. You know I would be walking around the track he would come next to me and say I think you ought to stop now. And I would say well why? Well your right leg is dragging a little bit but I didn't know that. Or he will, I'll have ten minutes set on a machine and he will say let's just stop at five and he really read me in terms of fatigue and um uneven heartbeat."

*Supportive energetic environment with opportunities for socialization and connection*: All participants commented positively on the environment. Participants described the environment as welcoming to all regardless of age or abilities. Three participants (25%) noted a comfort in knowing that those around them had experienced something similar. Additionally, three (25%) were inspired by the effort of those around them.

Participant 3: "Most people were very concentrated on what they were doing. A few people would say hello, but that was kind of it. And we were just all concentrating on what we were

doing. And it was kind of nice to see the level of energy and the people were working so hard. And that was kind of inspirational."

Participant 19: "Um, gym atmosphere is very, very loose. Um, it looks like the participants all understand the personalities of the different people working here. And so, I think it puts them at ease to be here because the age group of the people that are here are all over the place you have people that could be in their twenties to people that could be in their eighties and I've seen both ends of the spectrum and I've seen both, both sets of people very comfortable in what they're doing."

One contrary opinion on the gym environment came from a participant with sensory sensitivities secondary to her stroke. She reported difficulty, but also how she was able to overcome the barriers to participate.

Participant 11: "The only thing I would say specifically is, is the like I said, the conditions for stroke people and you know, I guess different strokes might have different needs. But, but the lights and sounds, that kind of thing, that surprised me that that was like a really big thing for me."

Participant 11: ". . .So, um I, when I would come, I would wear my sunglasses and I would put earbuds in and sometimes I would put earbuds in with no, with no music just to block out the sound. And I realized that if I did that, it could, it would calm me down and I could function."

Socialization opportunities varied from casual interactions to connections and friendships. Six participants (50%) noted casual, encouraging conversation with others while at CR. Several of the same participants and others (58%) noted opportunities for deeper connections because of shared experiences and re-connecting with old friends or making new ones.

Participant 25 (non-completer): "I don't remember anybody's names that I talked to. The one we, they'd walk like my speed around the track or whatever. But there'd be people that I'd see that we just kinda, like, clicked, just from seein' each other, right? Or we'd be workin' next to each other. . .it was just talkin' about regular things in life."

Participant 18: "I feel like people were just here doing their best. And that was good enough for me. Like there's. . . that vibe makes me feel like that's where I want to, uh, be in. And I-I- I have a home [exercise routine], but I like being around people. There's something about knowing other people are dealing with looking struggle in the face, and you're in a camaraderie about that."

## Safety

One serious safety event occurred during CR. One participant with a known atrial fibrillation diagnosis had an episode with a new rhythm (Fig 1). The CR staff put the participant on hold until she had permission from her cardiologist to return. CR staff contacted the cardiologist's office (outside of the health system) with information on the episode, and the participant returned to the program two weeks later, after an appointment with her cardiologist and a medication change.

Three serious safety events occurred outside of CR during the program period (one car accident, two medical complications) and caused temporary absence from the program for two participants, and inability to complete the program for one participant.

Non-serious events are presented in Table 4. Pain complaints occurred 45 times (7.4%) aggregated over all sessions for all participants. The average pre-session pain on a scale of 0–10 was 0.51(SD 0.87) with a range 0–9. The average post-session pain was 0.43 (SD 0.94) with a range of 0–8.

Consultations occurred between the PI and EP due to mobility concerns (n = 4) and safety concerns (n = 4). Mobility consultations included three study eligibility evaluations (both the PI and primary EP) related to cognitive issues (n = 1) and assistance required for participants with hemiplegia getting on and off equipment (n = 2). One additional mobility consultation occurred during the program between the PI and EP to address progression for a participant with gait and strength deficits (n = 1). The PI consulted with the primary EP on the four safety issues occurring during and outside of the CR program. The PI reported all safety events with physician visits to the health system and university IRBs; neither IRB considered any as sentinel events.

Safety themes from qualitative evaluation include regular monitoring and staff attentiveness promoting feelings of safety, and participants' perceptions of impairments impacting activity safety.

*Regular monitoring and staff attentiveness promoting feelings of safety*: Eight participants (67%) explicitly stated they never felt unsafe during the program. For most participants that sense of safety was due to the monitoring and staff attentiveness. Participants noted the staff was regularly focused on issues with blood pressure or heart rate (high or low) and responded quickly to atrial fibrillation episodes that the participants themselves did not recognize as anything problematic. Staff evaluated irregular heart rhythms, gave clear instructions to participants on findings, and contacted medical providers with detailed information.

**Table 4. Non-serious safety events.**

| Non-Serious Safety Event | Number of episodes recorded across all participants (% of 606 total sessions) |
|---|---|
| Falls at CR without injury | 1[a] (0.17%) |
| Falls outside of CR without injury | 3 (0.50%) |
| Soreness | 12 (2.00%) |
| Pain | 45 (7.40%) |
| Numbness | 2 (0.33%) |
| Dizziness | 12 (2.00%) |
| Shortness of Breath | 4 (0.66%) |
| Low Blood Sugar | 1 (0.17%) |
| High Blood Pressure at start | 9 (1.50%) |
| Low Blood Pressure at start | 3 (0.50%) |
| Atrial Fibrillation | 4[b] (0.66%) |
| Arrythmia | 4 (0.66%) |

Abbreviation: CR, Cardiac Rehabilitation

[a] One participant had a fall without injury at the post-program evaluation appointment where a stroke-related spatial relations issue caused the participant to miss a chair and sit on the floor. Participant was evaluated by CR staff and continued post-assessment.

[b] One of these atrial fibrillation events occurred in a patient who had been previously undiagnosed. CR staff and participant spoke to health system cardiologist, medication was prescribed, and participant returned to CR program the same week.

Additionally, staff monitored heart rates and blood pressure after sessions and had participants wait, drink water, or relax to normalize before they released them to leave.

Participant 19: "I felt very comfortable that God forbid I lost my balance and fell over and hit my head, or if my blood pressure was too high or too low, I feel very confident in the ability of the people that work here to react because I've seen accidents where anyone, another participant that had fallen and they jump faster than a cricket jumps. They just all of a sudden converge to the person that fell and they are on top of it."

Participant 4: "And so, I called [head exercise physiologist] over because I wanted to explain to him that the machine I was on was broken. Because it's reading my heart rate as 165. Okay. And of course, he took my pulse and um, said there's nothing wrong with the machine would you please come and sit over here. And uh, that's the very first time I was aware that I had had an a-fib and what an a-fib was. And um, (clearing throat) [head exercise physiologist] and [nurse], they both and everyone else spent all of the next forty-five minutes taking care of me."

Participant 3: "I think it was the checking of my heart beat and then using the strip and they were showing AFib and then really kind of wild variation. . . . Um, and so I called my doctor and got an appointment. . .. And he made a minor medication change for me, and I think it's really helped. So, my blood pressure is more predictable, and it's not crazy. And the dizziness is somewhat better. Um, so this was like a great service that this program did for me is to help me figure out that I needed some more attention and got it."

Participant 15: "And then at the end of the day there were three or four days that uh, I had to drink uh volumes of water and eat crackers before they would release me and uh I felt that was uh caring and thoughtful and although it was frustrating to, um, not be able to uh to just get on the way."

*Participant's perceptions of impairments impacting activity safety*: Three participants (25%) had safety concerns about specific activities at CR. One felt the treadmill and the rower were not safe because of her leg weakness. Another attributed a fall without injury at study post-assessment as related to stroke proprioceptive and cognitive processing deficits. The same participant worried about getting on and off the treadmill as well. Finally, another participant's dizziness, headache, vision symptoms, and knee pain made him feel less safe on the equipment.

Participant 3: "I have these stroke related things that are somewhat subtle but made it really impossible for me to do um the treadmill. That I would just fall. I didn't. I mean [head exercise physiologist] was with me, and I didn't fall but it was such a risk and then rowing machine I also had a hard time getting up and getting down."

## Discussion

CR programs using Medicare guidelines for dosage and components is a feasible, safe, and enjoyable exercise opportunity for survivors of stroke. Results support integrating survivors of stroke into existing programs. Participants were able to achieve exercise intensity, meeting RPE ranges throughout the program. CR provided the attention of qualified staff and had the accountability of a regularly scheduled program with the extra benefit of session time

flexibility. Recruitment and uptake were barriers to implementation of CR for survivors of stroke. These barriers may be mitigated by strategies to increase survivors' self-efficacy for exercise, to make referral easier for clinical providers and to reduce participant level barriers to participation.

## Survivors can integrate into current CR program structure

There are several studies and programs outside the U.S. that support the use of CR for stroke survivors, but they either lack the same dosage or create an entirely new program just for survivors [29]. While other research has examined modified CR in the U.S. and found it to be effective and feasible, these studies have not presented why programs separate from the standard CR programs are required or desired [46, 47]. Utilizing existing CR programs, which are widely available in the U.S., has potential implementation advantages over creating new programs. The current study demonstrates that stroke survivors with a variety of mobility and other limitations can meet CR standards with prescribed intensities, and that they value the variety of activities available. CR staff (EPs and nurses) are qualified to monitor the cardiovascular system as was well demonstrated in the current study. PT touchpoints in the current study were at referral, at initial evaluation for program modification, and during the program for consultation. The current study results support the use of standard CR staff personnel with PT support needed only for referral, consultation, and staff training, much of which could be handled through referral specification, CR staff training, and email or phone consultations.

## Survivors can meet CR RPE intensity demands

In the current study, CR improved cardiovascular endurance with progressive, moderate-intensity exercise adjusted to the individual, with staff providing motivation and expert monitoring. Survivors of stroke were able to meet the RPE intensity demands set by the program and met progressive HR range targets 58% of the time. Dosing at three times a week for 12 weeks was acceptable to most participants. These findings support standard CR model dosing for stroke survivors. Previous research and evidence-based recommendations also support CR dosing for survivors [48].

## CR is a safe environment for survivors to exercise

Staff monitored vitals, identified and addressed adverse cardiac events which ensured participant safety. Simple interventions such as water or nutrition and retesting of vitals allowed participants to continue with their exercise routine without interruption. Several of the issues efficiently identified and handled in the CR environment, including atrial fibrillation episodes (both new and chronic) and abnormal blood pressure or blood sugar readings would not have been monitored in a standard fitness facility. As a result, the CR facility promoted safety and helped address underlying medical issues that may affect exercise tolerance, safety and risk [49]. The participants recognized the focus on safety and the staff's expertise as major benefits of the program. Safe exercise for a population with cardiovascular co-morbidities like the current sample was a benefit of CR and supports use of CR as a transitional program between standard rehabilitation and community exercise for stroke survivors. Consultations related to mobility with the study PI, who provided mobility expertise as a PT, were very few and minor. The consultations that occurred during this study could likely be handled with a phone consult with a PT in standard practice in the future. Future studies will be needed to determine processes for this type of consultation before CR for survivors of stroke can become standard practice.

## Positive exercise beliefs promote CR participation for survivors

Program completers in the present study were an intrinsically motivated group of primarily previous exercisers. Program participation facilitators included previous exercise experience, belief that exercise is beneficial for health, flexible session times, making exercise a priority, and social support. These facilitators are consistent with the existing literature for traditional CR participants and the general exercise literature for survivors of stroke [50–53]. Program completers knew exercise was important for their health, wished to avoid further strokes and health complications, and wanted to improve themselves. The high motivation levels and exercise experience in this sample may be related to the importance of having self-efficacy and positive outcome expectations to commit to structured exercise programs [52, 54]. Despite these beliefs, many felt like they did not have the proper tools to exercise well on their own or lacked the accountability piece that the CR program provided. Social support encouraged continued participation through shared experiences with other participants, encouragement and ongoing evaluation from staff, and relationships with other participants and staff [50, 53, 55–58].

## Survivor recruitment and uptake is a challenge

Participant recruitment was the largest barrier to the present study. It is a common problem in traditional CR, with a 2018 report noting a 60–85% referral rate for common cardiac diagnoses and of those referred, a 50% uptake [59]. Referrals from the health system providers in the current study were low, and uptake for those referred from the health system was also low. Survivors of stroke were interested in the study because they self-referred through support groups and community members. An easy electronic referral process and prompts in electronic medical records to consider referral to these programs would reduce process burden on healthcare providers. Readily available educational materials for patients and training of healthcare providers could also influence referral and uptake [59, 60]. In one study by Anderson et al., stroke survivors preferred referrals for health system and community programs because they did not understand the rules for access or qualifications for specific programs [61]. This ambiguity was especially true for survivors with mild to moderate impairments [61]. In another survey-based study of 312 stroke survivors, the majority wished they performed more exercise, and 84% indicated interest in an exercise program if available [62]. Community outreach and education through support groups, a successful recruitment tool in the present study, is another potential recruitment tool that allows for education directly to the stroke survivor outside of the stressful health care environment [63].

## CR addresses many exercise barriers for survivors

Barriers to the regular participation in the program for study participants were related to other time obligations including work, transportation or distance to the site, and impairments related or unrelated to their stroke deficits. The CR program addressed several of these barriers through a variety of exercise activities that could be modified to accommodate mobility impairments, a staff qualified to address medical complications, and a flexible session schedule.

Educational interventions using evidence-based strategies may positively influence survivors of stroke with lower levels of exercise self-efficacy, a key driver of participation. These types of interventions have been shown to influence both stroke survivors and traditional CR participants to exercise [62, 64]. Utilizing self-efficacy for exercise outcome measures may help identify survivors of stroke at risk for not starting or not finishing programs in order to tailor interventions. PTs have a unique opportunity for these educational interventions and

previously mentioned referral interactions, because they have touchpoints with almost all survivors of stroke [65].

## Strengths, limitations and future directions

Strengths of the present study include application of standard U.S. based CR program dosage and intensity, including measures of intervention fidelity and an analysis of qualitative responses for participant program acceptability. Limitations in the present study include the evaluation of only one health system CR site in the Southeast U.S., limiting generalizations to other areas and programs. Additionally, participants did not take advantage of education sessions or consultation with psychologist which may have provided additional impact on outcomes. While the inclusion-exclusion criteria were broad with desires of reaching a varied sample, most of the pilot participants were white males who were already exercising and had few mobility limitations (self-selection bias).

Future research can expand on these findings through studies aimed at increasing participation by individuals with limited exercise experience, more women, and more diverse racial and ethnic backgrounds to better reflect the overall population of the community more directly. Additionally, more exploration of health system barriers to program referral and uptake and cost-effectiveness is suggested. Finally, further exploration of the impact and importance of weekly health education and further psychological consultation should be conducted.

## Conclusions

Survivors of stroke met CR program dosage and RPE intensity goals, and perceived the program as needed, regardless of their mobility limitations or previous exercise experience. Participants enjoyed the camaraderie and positive environment and felt safe and supported by staff. Challenges to CR integration included managing referral and uptake of the program. Clinical providers need an easy way to refer and educate patients on the importance of exercise. Survivors need positive exercise beliefs and strategies to overcome scheduling, transportation and other barriers. Findings support the use of CR programs for survivors of stroke after rehabilitation. Further investigations can confirm findings and explore integrating survivors of stroke as a standard of care, potentially impacting the health and mobility of a large number of survivors of stroke in the U.S.

## Supporting information

**S1 Checklist.**
(PDF)

**S1 File.**
(DOCX)

## Author Contributions

**Conceptualization:** Elizabeth W. Regan, Jill C. Stewart, Sara Wilcox, Stacy Fritz.

**Data curation:** Elizabeth W. Regan.

**Formal analysis:** Elizabeth W. Regan, Reed Handlery, Jill C. Stewart, Stacy Fritz.

**Funding acquisition:** Elizabeth W. Regan, Jill C. Stewart, Stacy Fritz.

**Investigation:** Elizabeth W. Regan, Jill C. Stewart, Stacy Fritz.

**Methodology:** Elizabeth W. Regan, Jill C. Stewart, Joseph L. Pearson, Sara Wilcox, Stacy Fritz.

**Project administration:** Elizabeth W. Regan, Stacy Fritz.

**Resources:** Elizabeth W. Regan.

**Software:** Elizabeth W. Regan.

**Supervision:** Elizabeth W. Regan, Jill C. Stewart, Sara Wilcox, Stacy Fritz.

**Validation:** Stacy Fritz.

**Visualization:** Stacy Fritz.

**Writing – original draft:** Elizabeth W. Regan, Stacy Fritz.

**Writing – review & editing:** Elizabeth W. Regan, Reed Handlery, Jill C. Stewart, Joseph L. Pearson, Sara Wilcox, Stacy Fritz.

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
