## [Decision Letter · Decision Letter 0]

30 Nov 2020

PONE-D-20-25410

Feasibility of integrating survivors of stroke into cardiac rehabilitation: a mixed methods pilot study

PLOS ONE

Dear Dr. Regan,

Thank you for submitting your manuscript to PLOS ONE. After careful consideration, we feel that it has merit but does not fully meet PLOS ONE’s publication criteria as it currently stands. Therefore, we invite you to submit a revised version of the manuscript that addresses the points raised during the review process.

We look forward to receiving your revised manuscript.

Kind regards,

Ukachukwu Okoroafor Abaraogu, BMR PT, MSc, PhD

Academic Editor

PLOS ONE

Journal Requirements:

2. In the methods section, please provide additional information to address the following: 1) the expertise and training of the interviewers and 2) where the study was conducted. and 3) please provide the complete clinical trial registration number.

Reviewers' comments:

Reviewer's Responses to Questions

**Comments to the Author**

1. Is the manuscript technically sound, and do the data support the conclusions?

Reviewer #1: Yes

Reviewer #2: Partly

Reviewer #3: Yes

2. Has the statistical analysis been performed appropriately and rigorously? 

Reviewer #1: N/A

Reviewer #2: No

Reviewer #3: No

3. Have the authors made all data underlying the findings in their manuscript fully available?

Reviewer #1: Yes

Reviewer #2: No

Reviewer #3: Yes

4. Is the manuscript presented in an intelligible fashion and written in standard English?

Reviewer #1: Yes

Reviewer #2: No

Reviewer #3: Yes

5. Review Comments to the Author

Reviewer #1: The authors have presented a feasibility study looking at integrating cardiac rehabilitation with stroke survivors. Overall i found this to be an interesting read and well-written albeit with a few mistakes that could have been picked up with proof-reading. I think this will be well received by readers and congratulate the authors for a well designed and delivered intervention. I have no major comments but some minor comments to be addressed.

Whilst acknowledging that the authors are based in the USA the introduction is very USA-centric, it would be useful to but the initial introduction as a more global context, what is the mortality, morbidity etc in a worldwide context.

I would encourage the authors to carefully proof read the article, there are several instances where they have put brackets and a word / XXX which has been left and not completed for example NCTXXXXX or OSF (OSF reference).

It may also be helpful to define what a CR programme looks like in the US for those who are unaware, do they differ from UK based programmes?

Please consider adding participant numbers into the results section rather than “a few” or “most participants”, so that we can get a sense of the % or number that are suggesting this. Is a few/most 3 or 12?

Page 3: Line 37 – Could the authors re-frame “as the insurance reimbursement climate changes”, I am unsure what these means.

Page 4: line 48 – Please provide a reference for widely available and well-established.

Page 4: Line 49 – Lack of consistency with abbreviations – cardiac rehab in full rather than CR

Page 4: Line 64 – Please include your NCT number rather than XXXX. Additionally please include your ethics number

Page 4: Line 66 – Assume OSF reference needs to be included here.

Page 5: Line 94 – please include an example(s) of modifications.

Page 5: Line 99 – Include Borg before scale

Page 5: Please confirm what target HR was for patients generally 60, 70, 80%? During the exercise component.

Page 7: Line 120 – (REF), either remove or include reference if appropriate.

Page 7: Line 122 – Please list your demographic information (i.e. age, sex, weight) and remove means, SD and ranges calculated.

Page 8: Line 136 – Incentive is not the correct terminology as it sounds coercive.

Page 8: Line 143/144 – Spelling error “in vivo” = NVIVO?

Page 18: Line 307/308 – If you are assessing HR and say that you calculated incorrectly, why do you not redo these calculations so that only the exercise component is included? If you still aren’t meeting HR in the sessions, this is still an important finding.

Page 18: Line 316-320 – I am unsure if this section on nutrition is appropriate to be placed in the adherence / fidelity section.

Page 23: Line 393 – Does this not contradict your previous statement about session times and days being available for patient selection?

Page 25: Line 435 – Contradicts page 13 “Barriers included medical complications, competing time demands, financial concerns, transportation difficulties (including long distance to site), and disinterest in gym exercise”

Page 30: I would caution against doing a statistical analysis as you are not powered to undertake this based on the fact it is a feasibility trial. You can show that there was an improvement, although it is only a difference of 61.9 metres with a large SD range. If you have formally powered for this analysis then this needs to be highlighted in the methods.

Page 31: Line 557 – Was it the right intensity based on the fact the HR was not correct?

Page 35: Note limitation should also include selection bias to those predisposed to take part in exercise, you have already mentioned this in your discussion so should be acknowledged as a limitation.

Page 35: You need to clarify if they did meet the intensity demands based on METS and HR, if you do a new calculation excluding warm-up and cool-down. If you don’t, you can say met based on estimated intensity from RPE score but not true intensity.

Reviewer #2: Authors sought to evaluate the feasibility of integrating stroke survivors into an existing cardiac rehabilitation (CR) program. To that effect, they aimed to answer key questions bothering on (1) recruitment, uptake, and retention, (2) adherence and fidelity, (3) acceptability, (4) safety, and (5) effectiveness.

The study methodology is lacking in depth, rigor, and particularity. Reflecting on one of the objectives of the study, the fidelity of the intervention, for example, It is unclear what an ideal CR program would be and the extent to which intervention delivery complied with a standard CR protocol. These are important considerations that require detailed clarification especially if findings must inform a larger trial. From the study, it appeared that an ideal CR program equates to exercise rehabilitation. But this is not clear in the current study.

Further, the reviewer feels that the question of the effectiveness of CR for stroke survivors was not sufficiently addressed by the study. One would ask why authors chose uncontrolled pre-post design [over a pilot RCT] for the quantitative arm of the mixed methods study given the inherent limitation of this design to address issues bothering on intervention efficacy/effectiveness. The reviewer also considers CR program to be a complex intervention that would require a minimum set of outcomes [or core outcome set] to justify intervention effectiveness. The prioritization of 6MWT as the only outcome to determine intervention effectiveness is grossly insufficient for a feasibility trial of CR program for stroke survivors.

The reviewer expected that authors would have published a comprehensive trial protocol in a peer-reviewed journal before the actual study to enable a rigorous and unbiased evaluation of core feasibility indices.

The authors need to describe the study methods and the results in different sections in the abstract. Otherwise, the paper was fairly written.

Reviewer #3: Thank you for inviting me to review this exciting manuscript entitled feasibility of integrating survivors of stroke into cardiac rehabilitation: a mixed-methods pilot study. This study examined the feasibility of integrating survivors of stroke into an existing CR. I commend the authors for taking this massive project and the ability to synthesis this project with multiple outcomes into a single mixed-method manuscript. Generally, the manuscript would improve if the authors address some of my concern below. In part, the manuscript was well written. The authors should expressly state their research questions and the corresponding study design that answered the question. For instance, from table 1, it seems that the qualitative part of the mixed method answered only recruitment, uptake and retention; acceptability and safety, why quantitative answers recruitment, uptake, retention; adherence and fidelity; safety and effectiveness. Since the effectiveness of the CR program has been reported elsewhere, can the authors consider only focusing on exploring only the program characteristics as it regards to recruitment, uptake, retention, adherence and fidelity, and safety?

See attached document for detailed review.

6. PLOS authors have the option to publish the peer review history of their article (what does this mean?). If published, this will include your full peer review and any attached files.

Reviewer #1: No

Reviewer #2: No

Reviewer #3: No

---

## [Author Response · Author response to Decision Letter 0]

14 Jan 2021

Dear Editors and Reviewers,

Thank you for your detailed review of our manuscript. We believe we have addressed all the included concerns and have a better manuscript as a result. The line numbers in the following reply to reviewer’s comments are the line numbers in the accepted changes version of the revised manuscript.

Reviewer #1: The authors have presented a feasibility study looking at integrating cardiac rehabilitation with stroke survivors. Overall i found this to be an interesting read and well-written albeit with a few mistakes that could have been picked up with proof-reading. I think this will be well received by readers and congratulate the authors for a well designed and delivered intervention. I have no major comments but some minor comments to be addressed.

Whilst acknowledging that the authors are based in the USA the introduction is very USA-centric, it would be useful to but the initial introduction as a more global context, what is the mortality, morbidity etc in a worldwide context.

The authors appreciate the need to have a more global focus at PLOs One and have added worldwide statistics to the introduction.

Line 25-26: More than 80 million people in the world are living after stroke with 13.7 million new cases each year.(1)

I would encourage the authors to carefully proof read the article, there are several instances where they have put brackets and a word / XXX which has been left and not completed for example NCTXXXXX or OSF (OSF reference).

The authors placed brackets and incomplete information to allow for anonymous review. These notations have been updated in the current version of the manuscript. 

It may also be helpful to define what a CR programme looks like in the US for those who are unaware, do they differ from UK based programmes?

The authors added details of CR program requirements in the U.S, and more details about studies in other countries including the U.K.

Line 66-69: “U.S. Medicare guidelines require CR programs to include exercise, risk modification intervention, psychosocial evaluation, and outcomes assessment within an individualized patient plan.(26) A maximum of 36 sessions within 36 weeks is the basic allowance, with at least 31 minutes of active exercise each session.(26) “

Line 54-60: “One example is a program in Canada that provided a stroke-specific program for survivors of stroke who have remaining mobility deficits post-rehabilitation and integrated those without mobility deficits into a traditional CR program.(29) The stroke-specific program was a once weekly 90 minute class including aerobic exercise, resistance training and health education.(29) Another program in the United Kingdom was a six-week program including 60-minutes of exercise, 60-minutes of education and 10-minutes of relaxation for participants after transient ischemic attack (TIA).(26)”

Please consider adding participant numbers into the results section rather than “a few” or “most participants”, so that we can get a sense of the % or number that are suggesting this. Is a few/most 3 or 12?

Authors have added number of participants / % of participants (12 participated in qualitative) to the theme notations throughout the results. 

Page 3: Line 37 – Could the authors re-frame “as the insurance reimbursement climate changes”, I am unsure what these means.

In the U.S., Medicare and other insurance companies are reducing reimbursement for inpatient rehabilitation services, reducing overall length of stay. Re-worded the sentence to the following for clarity:

Line 40-42: “Rehabilitation stays in the U.S. are declining in length due to reduced insurance reimbursement, potentially compounding the deconditioning remaining when rehabilitation is complete.(13, 14)”

Page 4: line 48 – Please provide a reference for widely available and well-established.

Line 50-51: Two references were added, Curnier 2005 which reviews the geographic distribution and number of program in the U.S (widely available), and Leon 2005 which is the American Heart Association’s recommendation for cardiac rehab programs and notes the first recommendations dating back to 1994 (well-established). 

Page 4: Line 49 – Lack of consistency with abbreviations – cardiac rehab in full rather than CR

The authors have updated to reflect CR instead of Cardiac rehab in this line and elsewhere. 

Page 4: Line 64 – Please include your NCT number rather than XXXX. Additionally please include your ethics number

The information was initially blinded for review. It has been updated.

Line 79-81: “The study was a registered clinical trial through the United States National Library of Medicine (ClinicalTrials.gov ID: NCT03706105). The health system Institutional Review Board (IRB) approved this study (18-1001), and the University of South Carolina IRB (Pro00079131) acknowledged it.

Page 4: Line 66 – Assume OSF reference needs to be included here.

The information was initially blinded for review. It has been updated with a reference.

Page 5: Line 94 – please include an example(s) of modifications.

The authors updated to reflect potential modifications. Actual modifications are provided in the results.

Method Update Line 112-114: “The primary investigator (PI), a licensed physical therapist, performed the screen to verify eligibility and determine any modifications in (i.e. equipment use, movement limitations, etc.) required for CR equipment or activities.”

Results related text line 250-255: . “Fifteen participants received recommendations for modifications to the CR program. Recommendations were primarily related to limitations or cautions on treadmill use or track-walking with an assistive device only. There were a few limitations of upper extremity overhead motion due to shoulder pain. Other recommendations were safety related, such as expressive speech limitations requiring visual cues for use of the RPE scale, and the presence of orthostatic hypotension requiring slow transitions.”

Page 5: Line 99 – Include Borg before scale

The line has been updated with “Borg”.

Page 5: Please confirm what target HR was for patients generally 60, 70, 80%? During the exercise component.

The information was updated to reflect the target heart rate levels.

Line 126-127: “Target heart rate (HR) was estimated from resting HR and HR at the completion of the 6MWT at a level between 60-85% of heart rate reserve.”

Page 7: Line 120 – (REF), either remove or include reference if appropriate.

The (REF) was removed.

Page 7: Line 122 – Please list your demographic information (i.e. age, sex, weight) and remove means, SD and ranges calculated.

The authors updated as requested. 

Line 157-159: “Demographic data was collected on all participants starting the program using a standardized intake form including age, gender, race/ethnicity, date of last stroke, stroke type, work status and exercise frequency.”

Page 8: Line 136 – Incentive is not the correct terminology as it sounds coercive.

The participants received gift cards for qualitative participation, the language has been updated to reduce impression of coercion. 

Line 173-174: “ Those participating completed informed consent and received a $20 gift card in appreciation of participation.”

Page 8: Line 143/144 – Spelling error “in vivo” = NVIVO?

The use of in vivo is correct in this context related to coding closely to the participants wording. The definition from sage publications research methods is: “In vivo coding is the practice of assigning a label to a section of data, such as an interview transcript, using a word or short phrase taken from that section of the data.” (https://methods.sagepub.com/reference/sage-encyc-qualitative-research-methods/n240.xml)

Page 18: Line 307/308 – If you are assessing HR and say that you calculated incorrectly, why do you not redo these calculations so that only the exercise component is included? If you still aren’t meeting HR in the sessions, this is still an important finding.

The datasheets available from the CR site do not notate the warm up and cool down periods separately. The data provided is heart rate notation on each piece of equipment and does not note which piece of equipment was used for warm-up and cool down specifically. 

Methods were updated to reflect when the HR and RPE were recorded for clarity

Line 137-138: “HR and RPE were recorded for each endurance activity during the session.”

Page 18: Line 316-320 – I am unsure if this section on nutrition is appropriate to be placed in the adherence / fidelity section.

Part of monitoring fidelity of the program was the program components. Tracking consultations related to nutrition and exercise was part of that and is listed in table 1 under fidelity. See below from table 1.

Adherence and Fidelity • Average number of sessions 

• Total completing at least 18 of 36 visits a 

• Average nutrition and exercise consultations

• Number attending weekly education sessions

• Number consulting psychologist 

• Frequency of each exercise activity (% of sessions)

• Frequency of optional activity (% of sessions)

• Average session exercise minutes (> 31 minutes) a

• Average exercise minutes spent at target intensities

Page 23: Line 393 – Does this not contradict your previous statement about session times and days being available for patient selection?

The previous statement noted that patients chose their time based on their preferences and program availability. This line was updated to indicate the specific program availability.

Line 131-133: “All standard program participants chose their days and times based on their schedule and program availability (Monday, Wednesday, Thursday 6am – 6pm, Friday 6am-12pm).”

Page 25: Line 435 – Contradicts page 13 “Barriers included medical complications, competing time demands, financial concerns, transportation difficulties (including long distance to site), and disinterest in gym exercise”

The barrier of disinterest was related to the gym machines or exercise itself. The theme of supportive energetic gym environment was about the energy and support of the people in the CR facility. Authors have updated to remove the word “gym” from the environment theme to reduce confusion. 

Line 487-491: “Supportive energetic environment with opportunities for socialization and connection: All participants commented positively on the environment. Participants described the environment as welcoming to all regardless of age or abilities. Others noted a comfort in knowing that those around them had experienced something similar. Some were inspired by the effort of those around them.”

Page 30: I would caution against doing a statistical analysis as you are not powered to undertake this based on the fact it is a feasibility trial. You can show that there was an improvement, although it is only a difference of 61.9 metres with a large SD range. If you have formally powered for this analysis then this needs to be highlighted in the methods.

The full participant outcomes and the related power analysis was detailed in a separate manuscript, now in press with the Journal of the American Heart Association. Reference to be provided once fully published. This manuscript provided in review as a related manuscript. Of note, the change was nearly double the minimal detectable change on the 6MWT for survivors of stroke (34m) with an effect size of 0.94. 

This section was updated to notate the published manuscript:

Line 607-609: Participant effectiveness published elsewhere (JAHA reference, when available in press), was primarily measured by cardiovascular endurance through the 6MWT which improved on average by 6.19m (p<0.0001, 95% CI 33.99 – 89.84m).

Page 31: Line 557 – Was it the right intensity based on the fact the HR was not correct?

Authors have updated the discussion to clarify how participants have achieved intensity goals.

Line 613-614: “Participants were able to achieve exercise intensity, meeting RPE ranges throughout the program.”

Line 638-639: “Survivors of stroke were able to meet the RPE intensity demands set by the program and meet progressive HR range targets 58% of the time.”

Page 35: Note limitation should also include selection bias to those predisposed to take part in exercise, you have already mentioned this in your discussion so should be acknowledged as a limitation.

Yes, self-selection bias is a problem for most community programs. Added a notation to limitations to note that the sample was limited due to self-selection bias. 

Line 712-714: “While the inclusion-exclusion criteria were broad with desires of reaching a varied sample, most of the pilot participants were white males who were already exercising and had few mobility limitations (self-selection bias).”

Page 35: You need to clarify if they did meet the intensity demands based on METS and HR, if you do a new calculation excluding warm-up and cool-down. If you don’t, you can say met based on estimated intensity from RPE score but not true intensity.

The conclusion was updated to reflect the RPE intensity goals were met.

Line 722-723: “Survivors of stroke met CR program dosage and RPE intensity goals, and perceived the program as needed, regardless of their mobility limitations or previous exercise experience.”

Reviewer #2: Authors sought to evaluate the feasibility of integrating stroke survivors into an existing cardiac rehabilitation (CR) program. To that effect, they aimed to answer key questions bothering on (1) recruitment, uptake, and retention, (2) adherence and fidelity, (3) acceptability, (4) safety, and (5) effectiveness.

The study methodology is lacking in depth, rigor, and particularity. Reflecting on one of the objectives of the study, the fidelity of the intervention, for example, It is unclear what an ideal CR program would be and the extent to which intervention delivery complied with a standard CR protocol. These are important considerations that require detailed clarification especially if findings must inform a larger trial. From the study, it appeared that an ideal CR program equates to exercise rehabilitation. But this is not clear in the current study.

The authors appreciate the comments regarding the components related to moving forward to further trials. Participants did not take advantage of the optional education sessions or the psychological consults which are part of the CR program. This was not clearly specified in the results, and a notation has been added to results and to limitations/ next steps. However, they did benefit from more than just exercise, through medical monitoring, nutrition consultation, and relaxation sessions in addition to the exercise component.

The following was added to clarify:

Results:

LINE 360-361: “No study participants attended available and optional weekly education sessions or psychologist consultation. “

Limitations:

Line 710-711: “ Additionally, participants did not take advantage of education sessions or consultation with psychologist which may have provided additional impact on outcomes.”

Line 719-720: “Finally, further exploration of the impact and importance of weekly health education and further psychological consultation should be conducted.”

Further, the reviewer feels that the question of the effectiveness of CR for stroke survivors was not sufficiently addressed by the study. One would ask why authors chose uncontrolled pre-post design [over a pilot RCT] for the quantitative arm of the mixed methods study given the inherent limitation of this design to address issues bothering on intervention efficacy/effectiveness. The reviewer also considers CR program to be a complex intervention that would require a minimum set of outcomes [or core outcome set] to justify intervention effectiveness. The prioritization of 6MWT as the only outcome to determine intervention effectiveness is grossly insufficient for a feasibility trial of CR program for stroke survivors.

The reviewer expected that authors would have published a comprehensive trial protocol in a peer-reviewed journal before the actual study to enable a rigorous and unbiased evaluation of core feasibility indices.

The authors appreciate that a RCT is a more robust design for efficacy. However, the authors decided on a pilot pre-post design in order to stage the eventual RCT well, which will also include a pre-published protocol. The current study’s single group phase I pilot study with feasibility analysis design will inform the next stages and is recommended as preparation for rehabilitation intervention randomized controlled trials. 

Related references:

1. Dobkin BH. Progressive staging of pilot studies to improve phase III trials for motor interventions. Neurorehabilitation and neural repair. 2009;23(3):197-206.

2. Orsmond GI, Cohn ES. The distinctive features of a feasibility study: Objectives and guiding questions. OTJR: occupation, participation and health. 2015;35(3):169-177.

3. Eldridge SM, Lancaster GA, Campbell MJ, et al. Defining feasibility and pilot studies in preparation for randomised controlled trials: development of a conceptual framework. PLoS One. 2016;11(3):e0150205.

The authors need to describe the study methods and the results in different sections in the abstract. Otherwise, the paper was fairly written.

Authors updated abstract to separate methods and results.

Reviewer #3:

Thank you for inviting me to review this exciting manuscript entitled feasibility of integrating survivors of stroke into cardiac rehabilitation: a mixed-methods pilot study. This study examined the feasibility of integrating survivors of stroke into an existing CR. I commend the authors for taking this massive project and the ability to synthesis this project with multiple outcomes into a single mixed-method manuscript. Generally, the manuscript would improve if the authors address some of my concern below. In part, the manuscript was well written. The authors should expressly state their research questions and the corresponding study design that answered the question. For instance, from table 1, it seems that the qualitative part of the mixed method answered only recruitment, uptake and retention; acceptability and safety, why quantitative answers recruitment, uptake, retention; adherence and fidelity; safety and effectiveness. Since the effectiveness of the CR program has been reported elsewhere, can the authors consider only focusing on exploring only the program characteristics as it regards to recruitment, uptake, retention, adherence and fidelity, and safety? I had conducted this review systematically using various headings, and I have provided examples (when necessary). See attached document for detailed review.

Title:

Consider adding- "Hospital-based" in the topic. e.g. Feasibility of integrating survivors of stroke into hospital-based, cardiac rehabilitation: a mixed-methods pilot study. 

The authors apologize for confusion related to the “hospital-based” notation, which in the U.S. implies a hospital system administered program in the outpatient setting. There are no true community-based CR programs in the U.S. There are also some differences now realized between U.S. and U.K phases. Authors have updated text to reflect the outpatient, medically monitored model. 

Line 70-71: “The purpose of this pilot intervention study was to examine the feasibility of integrating survivors of stroke into an existing, outpatient CR program in the southeastern U.S.”

Line 75-76: “The study was conducted at an outpatient CR site affiliated with a United States multi-hospital health system in the state of North Carolina.”

Line 82-85: “The health system CR program had an existing protocol for non-medicare covered diagnoses, primarily people with heart failure and cancer, to participate in the program modeled using a 12 -week structured and medically monitored outpatient CR protocol. (30) “

Abstract

• Can the authors consider separating methods from the result? 

Methods have been separated from results in the abstract 

• I would suggest that the authors revise the conclusion as it did not align with the result presented. Throughout the result section, there was no evidence about participants meeting the standard dosage and intensity goals. 

Conclusion has been updated for clarity related to dosage and intensity findings. Table 1 states the goal of minimum of 18 visits, and the medicare exercise minutes requirement (> 31 minutes has also been added to table 1 and in the text for clarity. 

Abstract conclusion line 19-21: “Conclusions: Survivors of stroke were able to meet Medicare standard dosage (frequency and session duration) and rate of perceived intensity goals, and perceived the program as needed regardless of their mobility limitations or previous exercise experience.” 

Introduction addition line 68-69: “ A maximum of 36 sessions within 36 weeks is the basic allowance, with at least 31 minutes of active exercise each session.(30) 

“

Introduction

• Line 27, can the authors explain a bit more the behaviour they were referring to in this statement. As it is written, the reader might not know the behaviour they were referring to. 

The behavior was referring to exercise training. Authors updated the sentence for increased clarity 

Line 29-31: “Studies support the feasibility and safety of exercise training for survivors of stroke and suggest that engaging in exercise can improve cardiovascular risk factors and endurance while reducing disabilities.(3-6)”

• Lines 36, can the authors be specific with the references. For instance, only 24% of time at > 40% maximum heart rate (MHR) in one (10), and 4.8% of the time at > 60% MHR in another (11).

Sentence updated with references after the information supported vs. end of sentence.

Line 39-40: “only 24% of time at > 40% maximum heart rate (MHR) in one study,(11) and 4.8% of time at > 60% MHR in another.( 12)” 

• The authors did well by mentioning studies primarily outside the US that implemented cardiac rehab programs for survivors of stroke. A summary of these studies and the differences with the authors' hospital-based CR would be useful. 

The authors have added more details on a few of the studies outside the U.S. highlighting the difficulties in applying to U.S. systems. Additionally, more clarity in what the U.S. guidelines and the specific protocols within those guidelines are provided.

Line 52-69: “Research studies, primarily outside of the U.S., have implemented CR programs exclusively for survivors of stroke and included survivors of stroke within cardiac-diagnosis specific programs.(24-29) One example is a program in Canada that provided a stroke-specific program for survivors of stroke with remaining mobility deficits post-rehabilitation and integrated those without mobility deficits into a traditional CR program.(29) The stroke-specific program was a once weekly 90 minute class including aerobic exercise, resistance training and health education.(29) Another program in the United Kingdom was a six-week program including 60-minutes of exercise, 60-minutes of education and 10-minutes of relaxation for participants after transient ischemic attack (TIA).(26) While these and other studies outside the United States support feasibility and benefit for survivors of stroke, several limitations exist. Limitations in existing studies include variation in dosage, limited evaluation of participant perception and limited external validity.(3-5) Additionally, it is difficult to apply these findings to U.S. settings due to differences in dosage, costs, and insurance coverage. As a result, a knowledge gap exists for the feasibility of CR for survivors of stroke within U.S. programs that follow Medicare guidelines.(30) U.S. Medicare guidelines require CR programs to include exercise, risk modification intervention, psychosocial evaluation, and outcomes assessment within an individualized patient plan.(30) A maximum of 36 sessions within 36 weeks is the basic allowance, with at least 31 minutes of active exercise required each session.(30) “

• Can the authors provide some distinction of community-based cardiac rehab programs and hospital-based (inpatient or outpatient) cardiac rehab programs?; this would provide a more robust rationale and justify why understanding the dosage and other parameters to ensure continuing of CR after discharge is essential. 

The authors apologize for confusion related to the community vs. hospital-based programs. There are no medicare covered truly community CR programs in the U.S., and the phases differ slightly from the U.S. to the world configurations. Wording was updated to reflect this program was an outpatient health system structured program (phase II or phase II/III depending on the source, so phases were omitted).

Line 82-85: “The health system CR program had an existing protocol for non-medicare covered diagnoses, primarily people with heart failure and cancer, to participate in the program modeled using a 12 -week structured and medically monitored outpatient CR protocol. (30) 

Methods

• Thank you for providing a reference for the health system CR. A brief description of the health system CR program, specifically on the Phases-as non-cardiac diagnoses were allowed to participate in the program after Phase II, would be helpful for the reader. 

The authors recognize that as an international journal, more detail is required regarding U.S. CR practices. Text was added to the methods to define phase II CR focus, and clarified that non-insurance covered diagnoses were included (cancer and heart failure primarily) when the study began. The authors believe the use of the word “after” to be confusing, and altered to “using”. 

Line 82-85: “The health system CR program had an existing protocol for non-medicare covered diagnoses, primarily people with heart failure and cancer, to participate in the program modeled using modeled using a 12 -week structured and medically monitored outpatient CR protocol.(30)”

• Rationales for inclusion criteria would be helpful or reference. For instance, why a diagnosis of stroke at least three months prior, ability to walk at least 40 meters with or without an assistive device etc. 

 The authors have updated the inclusion criteria to include rationales:

Line 90-101: “(1) a diagnosis of stroke at least 3 months prior to reduce the likelihood of outcomes being confounded by spontaneous recovery; 

(2) completion of physical and occupational therapy rehabilitation, if applicable to avoid confounding the results; 

(3) clearance by treating medical provider (physician or nurse practitioner) to participate as is standard CR program requirement; 

(4) ability to walk at least 40 meters with or without an assistive device as a CR program requirement related to staffing; 

(5) ability to transfer from sit to stand without external assistance as a CR program requirement related to staffing; and 

(6) ability to follow instructions and to communicate exertion, pain and distress as a safety measure.“

• Can the authors provide specifically the component of the standard CR and the specific modifications? As it is written in lines 92-109, the reader is not sure if it is a standard program or the modified program. 

Methods were updated to clarify what was standard practice and what was a modification to the CR program for the study.

Line 1190122: “The only differences between the standard CR program and the study program were (1) Individual equipment or activity modification recommendations, and (2) the availability of the PI for consultation (in person and by phone) to address any mobility impairment issues arising 

Line 124-148: “The standard exercise program began with an analysis by the Exercise Physiologist (EP) to determine baseline levels of exercise intensity in metabolic equivalents (METs) based on participant’s 6MWT results. (34, 35) Target heart rate (HR) was estimated from resting HR and HR at the completion of the 6MWT at a level between 60-85% of heart rate reserve. Target exercise rating of perceived exertion (RPE) levels were set from 11-14 (somewhat hard to hard) on a Borg scale of 6 - 20.(35, 36) Standard program sessions were scheduled three times a week for 12 weeks with a target of 31-50 minutes of cardiovascular endurance activity each session. All standard program participants chose their days and times based on their schedule and program availability (Monday, Wednesday, Thursday 6am – 6pm, Friday 6am-12noon). Training sessions were individualized for all standard program participants. While components varied by session and individual, the standard format was 8 to 10 minutes of warm up, 10 to 40 minutes of cardiovascular endurance activities (treadmill, recumbent step machine, recumbent bike, over ground walking) and strength-building (resistance exercises), and a 5 to 10 minute cool down followed by optional activities. Optional activities within the standard program included strengthening, stretching, and/or relaxation, depending on the individual needs and goals. Progression in the standard program was determined through participant response. If RPE was consistently rated < 11, METs level effort was increased to reach an RPE of 14. Regular standard monitoring was completed both pre- and post-session for blood pressure, HR, and as needed, blood sugar and heart rhythms. Discontinuation of a standard session occurred if blood pressure exceeded 170/100 or by clinical expertise of the EP or staff nurses. 

 Psychosocial and nutritional consultation were available to all standard participants and were offered to study participants with stroke as but were not required. The PI discussed these options with study participants at the initial evaluation, and study participants were instructed to discuss their interest with the primary EP. Interest was recorded on the plan of care.” 

• What does the standardised intake form consist of? This will align adequately when the authors described the demographics. For instance, the standardised intake form consists of the age, sex and number of PA activity before enrolling into the program, type of stroke, affected area of the stroke, etc. 

The authors updated as requested. 

Line 157-159: “Demographic data was collected on all participants starting the program using a standardized intake form including age, gender, race/ethnicity, date of last stroke, stroke type, work status and exercise frequency.”

• I would suggest the authors check the equator network to consider which reporting guideline is appropriate for their study. For instance, there should be a heading for a program description, which the authors described the health system CR program, its standard components and its modifications. I believe all of these are in part located in across the manuscript, but reporting guideline would help organise the information for an easy read. 

The CONSORT requirements were as close as possible to review this feasibility study and the COREQ requirements were reviewed for the qualitative portion. Copies are provided in the review response. 

• The authors should develop a qualitative methodology. What kind of qualitative paradigm, descriptive or interpretative. What is the criteria to participate in the qualitative criteria, was it criterion-based purposive sampling or convenience sampling.

The authors chose to take a pragmatic, general qualitative approach to evaluating the feasibility variables and outcomes desired as presented by Orsmond and Cohn. A pragmatic approach is a mixed methods study paradigm that aims to evaluate real-world experiences. The pragmatic focus is noted in the first paragraph of methods section and also under the sub-heading Qualitative Procedures and Analysis. 

Orsmond GI, Cohn ES. The distinctive features of a feasibility study: Objectives and guiding questions. OTJR: occupation, participation and health. 2015;35(3):169-177.

Kaushik V, Walsh CA. Pragmatism as a Research Paradigm and Its Implications for Social Work Research. Social Sciences. 2019; 8(9):255.

• In Table 1, did the authors consider asking a question about patients perception about the benefits of the program? 

Authors have presented the outcomes data in a separate manuscript that was provided in the review process. It is currently in press with the Journal of the American Heart Association and the details are presented in that manuscript on participant perceptions of benefits/outcomes. 

• What guideline was used in the structured observation? What exactly was the PI observing? It would be great to have this information for the reader to understand how it added to the qualitative rigour. Does this mean that data for the qualitative component of this feasibility pragmatic study were: single individual face-to-face or telephone interview and PI's observation? If yes, it should be stated clearly. Does the PI observation purely for triangulation or to provide for context to the individual interviews? These should be clear in the method section of the qualitative study.

The observation was for triangulation of data. The information to observe (while the PI was in the CR facility for other reasons), included viewing a participant, the activities they performed, the interactions with other individuals (staff, other stroke participants, standard CR participants), the context of the interaction (social, instructional etc), and barriers/facilitators to participation including any safety concerns. These topics are related to interview topics listed in table 1.

Line 178-181: “In addition to interviews, the PI completed monthly structured observations of participant activities, interactions with others, and barriers and facilitators including safety to provide supplementary data and to add to qualitative rigor.(40)”

• Was there any need for any form of member checking for the qualitative component? How do the authors' reflexivity influences data analysis? 

There was not a perceived need for member checking. While the PI had clear interest in success of the program, she was aware and worked to use reflection memos and discussions with other authors to reduce these influences where possible. The other authors had no direct contact or benefit to the results assisting in reducing any implicit bias. 

The following in methods was updated to reflect:

Line 190-191: “Discussions between coders and with research mentor (last author) supported refinement and attempted to reduce bias.”

• Generally, as I understand that the authors used both inductive and deductive thematic approaches to analyse their data, but they failed to state how they combined these two distinct approach. At what level was inductive occur and deductive? Does it mean the PI first round of open coding is considered inductive, and the use of codebook is considered deductive? Deductive, the two coders map the themes emerged under the broad categories. If yes, these should be very explicit in the data analysis method.w I would suggest authors look for frameworks for program uptake, barriers and facilitators to guide their deductive analysis. 

The authors have attempted to clarify the basis for coding procedures and where the general topics came from. 

Line 185-186: “A codebook was created with broad categories based on inductive findings and feasibility topics (Table 1) for recruitment, barriers and facilitators,”

Line 188-190: “Two researchers (the PI and second author) iteratively completed deductive coding using the codebook in three rounds.”

• Did the authors consider if the type of stroke (ischemic or hemorrhagic or others could have any impact on the 1) recruitment, uptake and retention, (2) adherence and fidelity, (3) acceptability, (4) safety, and (5) effectiveness? as they did not consider that in their method section. What is the severity of their stroke? Does this matter or not? This information will provide context to the study to ensure applicability in a similar context (external validity). 

Authors recorded type of stroke, but did not do any impact analysis as there were not enough of each type to get a true consideration. Authors felt the severity of mobility impacts was more pertinent and attempted to reflect that by presenting initial walking speed. A notation was made in the table 2 description of survivors of stroke > 0.8 m/s as unlimited community ambulators to give this context. 

Table 2 legend: “Survivors of stroke with self-selected walking speed of greater than 0.8 m/s are considered unlimited community ambulators.(43)”

Result:

• Table 2: Can the authors also provide a range for the time since stroke, as it will give us additional information? How was the pre-program exercise level calculated? This may be in the body of the text, but Tables should be a stand-alone, and a foot description of how it was calculated will have the reader? 

Table 2 has been updated to reflect the range for age and time since stroke for both categories. A notation in the legend reflects that pre-program exercise level was self-reported by participants. 

• Reporting individual deficits (lines 202-205) might not be the best approach. Can the authors consider reporting individuals with single deficits (weakness= 13, walking= 10..... ) and those with two or more deficits (n=). These will give us more information in terms of deficits. We know that multiple deficits, either physical, cognitive or environmental, can influence adherence. Did the authors consider this?

The authors appreciate the need to distinguish how many participants had multiple deficits, which was the majority. The combinations of deficits created too many categories to report. The authors did update to note the single deficits (n=5) and multiple deficits (n=19). 

The text was updated: 

Line 246-250: “Participants described remaining deficits related to their stroke as weakness (n=13), walking (n=10), balance (n=9), coordination (n=7), speech (n=6), cognitive (n=5), vision (n=4) and memory (n=4). Five participants reported single deficits: weakness, balance, speech, memory and sensation. The remaining 19 participants reported multiple deficits.”

• I believe that the authors quoted verbatim the participants responses in the manuscript. Providing grammatical corrections will improve the readability of the participant's quote. For instance, participant 4 (under competing time demands), can be written as: So [am] not being able to [be] here [attend the CR program] three times a week um, work came into play. Alternatively, better still choose a more clear quote to support any theme emerged. 

Authors appreciate the readability issue. However, making changes alters the voice of the participant and is undesired. Participant 4 had mild aphasia and using only his quotes that were easily read would cause missing his perspective. The authors prefer to maintain the voice the participants as they spoke and have left the quote as is. 

• Lines 245 -248 is confusing. The first sentence seems like the authors' perception. While the first sentence is significant, it would be great if the authors present the result and then provide their perspective as to why people enrolled in the program in the discussion section of the manuscript. 

Authors updated these lines to notate that the reasoning came from a participant noting the expense of personal training for clarity. It was not a perception by the author. 

Line 290-292: “Financial concerns: While most participants did not mention financial concerns, the no-cost factor facilitated three (25%) enrolling in the study. One participant noted that alternatives such as personal training were too expensive.”

• The theme "disinterest in gym exercise", is it more of disinterest in structured gym exercise...

Authors believe the theme is not about the structure but primarily the equipment or formal exercise vs. physical activity or just even exercise in general. The theme has been updated to reflect all of these issues more accurately to remove the word “gym”. 

Line 266-268: “Barriers included medical complications, competing time demands, financial concerns, transportation difficulties (including long distance to site), and disinterest in exercise.”

Line 298 (heading): “Disinterest in exercise:”

• Line 312, is it all participants that were informally exposed to nutrition and exercise consultation? 

Information on exercise and nutrition consultation during sessions did not have a standard area in documentation, so it was recorded if the supervising staff member wrote it in the notes section for the session. The data presented notes that these happened from 0-5 times (nutrition) and 0-4 times (exercise) based on the notes and presented in the lines following the one noted above. The results were updated to note that none of the study participants attended the education sessions and that education that did occur was informal and individualized.

Line 360-363: “No study participants attended available and optional weekly education sessions or psychologist consultation. Education of participants was informal, depended on the staff member and the participant, and included nutrition and exercise consultation and advice.”

• Based on table 1, there is no qualitative interview topic for adherence and fidelity. Can the authors align which area the qualitative or quantitative data answered? If both data answered all areas, the authors should also state it and then ensure that all outcomes for quantitative data and interview questions reflect in Table 1. 

Authors appreciate the detailed review of table 1. Barriers and facilitator questions also impacted adherence, so those topics were added to the qualitative area of table 1. 

• Thank you for providing the type of equipment the participants did not like. While this is informative for clinicians, did the participants stated why they prefer some equipment and why they did not prefer some. Authors should provide reasons why participants prefer one thing vs the other(see line 393), and throughout the manuscript. 

The authors added a line to this theme area reflecting the participant’s reasoning for like or dislike.

Line 396-398: “Reasoning for preferences were due to enjoyment of the equipment, their perceptions of their capabilities on the equipment or personal goals to work a body area that was the equipment focus.” 

• Thank you for providing the statement on line 366. This type of statement should be stated in the method section of the manuscript. 

Methods were updated to more clearly reflect in the text which variables were evaluated where. Details remain in table 1 which is referenced. The overall approach sentence was moved out of the quantitative analysis area to the general area, and new statements reflecting table 1 divisions were added to the quantitative and the qualitative portions of methods.

This was moved:

Line 152-155: “Process variables and feasibility measures were recorded and analyzed throughout the study (Fig 1). The following categories were analyzed for feasibility, with details on which areas were covered by quantitative or qualitative measures in Table 1: (1) recruitment, uptake and retention, (2) adherence and fidelity, (3) acceptability, (4) safety, and (5) effectiveness.”

This was added: 

Line 160-161:” Details of the quantitative process variables are presented in table 1 and include measures for recruitment, uptake and retention, adherence and fidelity, safety, and effectiveness.”

Line 167-170: “Table 1 presents key areas of evaluation based on a previous study evaluating CR for colorectal cancer (39) and aim to include participant perspectives on key pilot feasibility study characteristics including recruitment, uptake, retention, acceptability and safety.(32).”

• The three safety severe events occurred outside CR was not associated with the CR. if it is related to the CR, they should provide context on how it was related to the events. This is a situation a qualitative interview will provide context on how the three serious safety events are related to the CR program. I would advise authors only to report safety events that are related to the CR program. We cannot control what happens outside the CR program or can infer the impact of the CR during the CR program period. 

The authors added context to why reporting the serious safety events outside of CR was important (absence from the program).

Line 538-540: “Three serious safety events occurred outside of CR during the program period (one car accident, two medical complications) and caused temporary absence from the program for two participants, and inability to complete the program for one participant. “

• How is (a) the number and type of mobility impairment, and (b) mobility and safety consultations,- safety issues? Does it mean the participants developed these mobility impairments when they started the program or what? Can the authors provide context on this. How were the safety events assessed, self-reported or via observation as the author mentioned? 

A. The number and type of mobility impairments was reported to understand the participant’s level and existed prior to the start of the program. The authors believe it was misleading to include as part of the safety measures in table 1, and it was removed. The original intent was to record any safety issues that were related to mobility limitations, but due to the definition of serious and non-serious safety events, these were already covered. 

Table 1 update removed: number and type of mobility impairments from safety area. 

B. The mobility and safety consultations were recorded as a way to maintain safety within the research and also to help understand whether additional PT expertise might be required to manage survivors of stroke within traditional cardiac rehab. 

Methods were updated to reflect the importance providing this service for for study safety and discussion updated to reflect the impact on future studies and implementation as standard practice. 

Line 654-658: “Consultations related to mobility with the study PI, who provided mobility expertise as a PT, were very few and minor. The consultations that occurred during this study could likely be handled with a phone consult with a PT in standard practice in the future. Future studies will be needed to determine processes for this type of consultation before CR for survivors of stroke can become standard practice.”

Table 1 updated to note consultation over communication for mobility and safety for consistency in language.

Discussion 

• Line 557 stated on CR was the right intensity was not supported by the study found. Rather participants stated that they liked the frequency, and no question of intensity was asked. Can the authors consider rewording this statement?

Authors updated this line and added another in order to clarify the intensity findings.

Line 613-616: “Participants were able to achieve exercise intensity, meeting RPE ranges throughout the program. CR provided the attention of qualified staff and had the accountability of a regularly scheduled program with the extra benefit of session time flexibility.”

Line 638-639: ” Survivors of stroke were able to meet the RPE intensity demands set by the program and meet progressive HR range targets 58%

---

## [Decision Letter · Decision Letter 1]

3 Feb 2021

Feasibility of integrating survivors of stroke into cardiac rehabilitation: a mixed methods pilot study

PONE-D-20-25410R1

Dear Dr. Regan,

We’re pleased to inform you that your manuscript has been judged scientifically suitable for publication and will be formally accepted for publication once it meets all outstanding technical requirements.

Kind regards,

Ukachukwu Okoroafor Abaraogu, BMR PT, MSc, PhD

Academic Editor

PLOS ONE

Additional Editor Comments (optional):

Many thanks for comprehensively responding to the reviewers comments. Two reviewers have now recommended to accept while a third reviwer recommend minor revision. The 3rd reviwers raised still an issue about the suitability of including effectiveness data much so as this has been reported elsewhere (as stated by the authors), wanted to see important references added to coments in line 77, and recommended changes in the language used in lines 168. The third reviwers also suggested to include a statement that 'patients' perception of the program's benefits has been published elsewhere', with the reference to back it up.

I agree with the two reviewers that the authors have responded to all ealier querries. I also agree with the 3rd reviewer that the additional points raised are minor but important. I will go ahead to accept this manuscript but will want the author to effect the changes further suggested by the 3rd reviwer before the manuscript goes to production stage.

Reviewers' comments:

Reviewer's Responses to Questions

**Comments to the Author**

1. If the authors have adequately addressed your comments raised in a previous round of review and you feel that this manuscript is now acceptable for publication, you may indicate that here to bypass the “Comments to the Author” section, enter your conflict of interest statement in the “Confidential to Editor” section, and submit your "Accept" recommendation.

Reviewer #1: All comments have been addressed

Reviewer #2: All comments have been addressed

Reviewer #3: All comments have been addressed

2. Is the manuscript technically sound, and do the data support the conclusions?

Reviewer #1: Yes

Reviewer #2: Yes

Reviewer #3: Yes

3. Has the statistical analysis been performed appropriately and rigorously? 

Reviewer #1: N/A

Reviewer #2: Yes

Reviewer #3: Yes

4. Have the authors made all data underlying the findings in their manuscript fully available?

Reviewer #1: Yes

Reviewer #2: (No Response)

Reviewer #3: (No Response)

5. Is the manuscript presented in an intelligible fashion and written in standard English?

Reviewer #1: Yes

Reviewer #2: Yes

Reviewer #3: Yes

6. Review Comments to the Author

Reviewer #1: (No Response)

Reviewer #2: (No Response)

Reviewer #3: I must commend the authors for revising the manuscript. It has improved significantly. However, I have a few minor comments.

1. Generally, It has been agreed that feasibility or Pilot study may not be used to determine the effectiveness of an intervention.

See:

Arain, M., Campbell, M. J., Cooper, C. L., & Lancaster, G. A. (2010). What is a pilot or feasibility study? A review of current practice and editorial policy. BMC medical research methodology, 10(1), 1-7.

Whitehead, A. L., Sully, B. G., & Campbell, M. J. (2014). Pilot and feasibility studies: is there a difference from each other and a randomized controlled trial? Contemporary clinical trials, 38(1), 130-133.

Based on this and the fact that this study's effectiveness has been published elsewhere (JAHA reference). I would suggest that the authors remove the (5) of the study aim.

Specific:

2. Lines 77, can the authors provide the references (e.g. Orsmond and Cohn, 2015; Kaushik and Walsh, 2019)

3. Lines 168, I would recommend describing or evaluating, as the former is more of a qualitative language. Also, provide references as above.

3.After the statement at line 171, the authors could state that patients' perception of the program's benefits has been published elsewhere (reference). This is in regards to the article currently in press in the J. of American Heart Association.

7. PLOS authors have the option to publish the peer review history of their article (what does this mean?). If published, this will include your full peer review and any attached files.

Reviewer #1: **Yes: **Amy Elizabeth Harwood

Reviewer #2: **Yes: **Elochukwu Fortune Ezenwankwo, PT, MSc (Med), MPH

Reviewer #3: No

---

## [Editor Report · Acceptance letter]

18 Mar 2021

PONE-D-20-25410R1 

Feasibility of integrating survivors of stroke into cardiac rehabilitation: a mixed methods pilot study 

Dear Dr. Regan:

I'm pleased to inform you that your manuscript has been deemed suitable for publication in PLOS ONE. Congratulations! Your manuscript is now with our production department. 

Kind regards, 

on behalf of

Dr. Ukachukwu Okoroafor Abaraogu 

Academic Editor

PLOS ONE